# High-contrast "gaudy" images improve the training of deep neural network models of visual cortex

**Benjamin R. Cowley, Jonathan W. Pillow**
Princeton Neuroscience Institute, Princeton University
{bcowley, jpillow}@princeton.edu

## Abstract

A key challenge in understanding the sensory transformations of the visual system is to obtain a highly predictive model that maps natural images to neural responses. Deep neural networks (DNNs) provide a promising candidate for such a model. However, DNNs require orders of magnitude more training data than neuroscientists can collect because experimental recording time is severely limited. This motivates us to find images to train highly-predictive DNNs with as little training data as possible. We propose high-contrast, binarized versions of natural images—termed gaudy images—to efficiently train DNNs to predict higher-order visual cortical responses. In simulation experiments and analyses of real neural data, we find that training DNNs with gaudy images substantially reduces the number of training images needed to accurately predict responses to natural images. We also find that gaudy images, chosen *before* training, outperform images chosen *during* training by active learning algorithms. Thus, gaudy images overemphasize features of natural images that are the most important for efficiently training DNNs. We believe gaudy images will aid in the modeling of visual cortical neurons, potentially opening new scientific questions about visual processing.

## Introduction

A major goal in systems neuroscience is to understand the sensory transformations of the visual system [1, 2]. A key part of this goal is to find a model that accurately maps natural images to the responses of visual neurons. For lower-order visual areas, DNNs have been found to be highly predictive of responses from retinal cells [3–5] and neurons in primary visual cortical area V1 [6–10]. For higher-order visual areas V4 and IT (the focus of our work), the amount of data offered by neurophysiological experiments (where recording time is limited and costly) is often too small to train end-to-end DNNs without overfitting. To overcome this, transfer learning is typically used: Images are first passed into a DNN already trained for object recognition, and the activity of hidden units from a middle layer of this pre-trained DNN are then mapped to neural responses [11–14]. This mapping is almost always chosen to be linear, both for interpretability reasons and to avoid overfitting. However, the true mapping between features and responses is likely nonlinear. We desire to train a more expressive mapping: a readout network (i.e., a DNN) that can fit to nonlinear mappings. Because DNNs tend to overfit to a small number of training responses (even with transfer learning), **our goal is to optimize images to train the readout network as accurately as possible with as little training data as possible (i.e., minimize recording time).**

Here we report a surprising finding: A simple manipulation of natural images substantially reduces the number of training images needed. This manipulation, inspired by active learning theory, maximizes the dynamic range of each input dimension in order to drive the activity of hidden units in the readout network as much as possible. It achieves this by setting the pixel intensity $p$ of each color channel to either 0 or 255, depending on if $p$ is less than or greater than the image's mean pixel intensity,

respectively. We refer to the resulting high-contrast, binarized natural images as "gaudy" images for their flashy bright colors and strong local contrasts. In extensive simulation experiments and analyses of real V4 neural data, we find that gaudy images reduce the number of images needed to train readout networks with different activation functions, different numbers of layers, and different architectures, as well as for different pre-trained DNNs used to simulate visual cortical responses. The success of gaudy images comes from their high training errors and from their overemphasis of high-contrast edges. In addition, we find that gaudy images, generated *before* training, lead to performances on par with or greater than those achieved by images chosen *during* training by active learning algorithms [15, 16]. This suggests that gaudy images overemphasize the features of natural images (e.g., high-contrast edges) that are most important to efficiently train the readout network—features that active learning algorithms must find without explicit guidance.

Our results are likely to be of broad interest. Visual neuroscientists can test the efficacy of gaudy images to train models to predict visual cortical responses, and it is an open scientific question whether visual neurons respond differently to gaudy images versus normal natural images. In addition, improving models of visual cortical neurons will improve methods that rely on the predictions of these models, such as adaptive stimulus selection techniques for optimizing neural responses [17–23]. Researchers studying the similarities and differences of image representations between two different DNNs trained for object recognition [24–26], between a pre-trained DNN and neural responses [27, 28], or between a pre-trained DNN and human/animal perceptual behavior [29] may also benefit from using gaudy images to probe these representations. In general, gaudy images add to the growing number of techniques, including transfer learning and data augmentation, that general practitioners may use to more efficiently train DNNs in data-limited regimes.

# 1 Gaudy images are inspired by active learning theory.

What is the optimal set of images to train DNNs with as little training data as possible? To begin to answer this question, we recall a counterintuitive theoretical result from active learning (AL) and optimal experimental design: In the case of a linear relationship between stimulus and response, the optimal strategy for fitting a linear model is to choose stimuli *before* collecting responses or training the model [30, 31]—contrary to the strategy of the typical AL algorithm, which chooses stimuli *during* model training. The optimally-chosen stimuli are outliers of the dataset (i.e., stimuli that maximize the variance of the input variables). We first give the mathematical underpinnings for this theory and then propose gaudy images as instances of such optimal stimuli.

Formally, let us assume a linear mapping between vectorized image $\mathbf{x} \in \mathbb{R}^K$ (for $K$ pixels) and response $y$: $y = \beta^T \mathbf{x} + \epsilon$, with weight vector $\beta \in \mathbb{R}^K$ and noise $\epsilon \sim \mathcal{N}(0, \sigma_{\text{noise}}^2)$. We model this mapping with $\hat{y} = \hat{\beta}^T \mathbf{x}$, where $\hat{\beta} = \Sigma^{-1} \mathbf{X} \mathbf{y}$ for a set of (re-centered) training images $\mathbf{X} \in \mathcal{R}^{K \times N}$ (for $K$ pixels and $N$ images), unnormalized covariance matrix $\Sigma = \mathbf{X} \mathbf{X}^T$, and responses $\mathbf{y} \in \mathbb{R}^N$. Consider the expected error that represents how well our estimate $\hat{\beta}$ matches that of ground truth $\beta$: $E[\|\beta - \hat{\beta}\|_2^2]$. Given previously-shown images $\mathbf{X}$, we seek a new, unshown image $\mathbf{x}_{\text{next}}$ from the set of all possible (re-centered) images $\mathcal{X}$ that minimizes this error after training $\hat{\beta}$ on $\mathbf{x}_{\text{next}}$ and its response $y_{\text{next}}$. Importantly, we do not know $y_{\text{next}}$, so we cannot ask which image has the largest prediction error $\|y_{\text{next}} - \hat{y}(\mathbf{x}_{\text{next}})\|_2^2$. Instead, we choose the image $\mathbf{x}_{\text{next}}$ that minimizes the error between the true $\beta$ and our estimate $\hat{\beta}$ trained on images $[\mathbf{X}, \mathbf{x}_{\text{next}}] \in \mathbb{R}^{K \times (N+1)}$ (see the Appendix for full derivation):

$$\mathbf{x}_{\text{next}} = \underset{\mathbf{x} \in \mathcal{X}}{\arg\min} \, \mathrm{E}\left[\|\beta - \hat{\beta}\|_2^2 \mid [\mathbf{X}, \mathbf{x}]\right] = \underset{\mathbf{x}}{\arg\max} \, \frac{\mathbf{x}^T (\Sigma^{-1})^2 \mathbf{x}}{1 + \mathbf{x}^T \Sigma^{-1} \mathbf{x}} \tag{1}$$

We can maximize this objective by choosing the image $\mathbf{x}$ with the largest projection magnitude along the top eigenvectors of $\Sigma^{-1}$. To better intuit this optimization, let us assume that the $K$ pixels of $\mathbf{X}$ are uncorrelated (e.g., via a change of basis), resulting in the covariance matrix of pixel intensities $\Sigma$ to be diagonal with entries $\Sigma_{k,k} = \sigma_k^2$. Under this assumption, the optimization becomes the following:

$$\mathbf{x}_{\text{next}} = \underset{\mathbf{x} \in \mathcal{X}}{\arg\max} \, \frac{\mathbf{x}^T (\Sigma^{-1})^2 \mathbf{x}}{1 + \mathbf{x}^T \Sigma^{-1} \mathbf{x}} \stackrel{\Sigma = \text{diag}(\Sigma)}{=} \underset{\mathbf{x}}{\arg\max} \sum_{k=1}^{K} \mathbf{x}_k^2 / \sigma_k^2$$

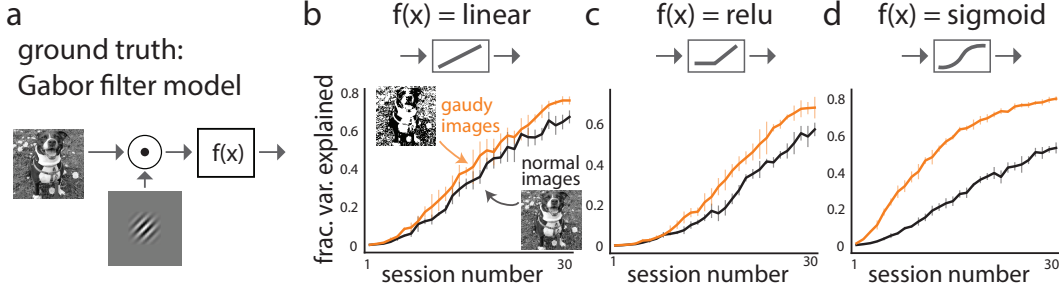

**Figure 1: Gaudy images efficiently train generalized linear models (GLMs). a**. For ground truth, we use a Gabor filter model with activation function $f(x)$ with no noise added to the output (adding output noise yields similar results). **b**. Training the filter weights of a GLM to predict ground truth responses. Fraction of variance explained ($R^2$) is computed from responses to heldout normal images only. The $f(x)$ for both GLM and ground truth is linear. Grayscale gaudy images have pixel intensities of either 0 or 255 (see text). **c**. Same as **b** except $f(x)$ is a relu for both. **d**. Same as **b** except $f(x)$ is a sigmoid for both. Performance plateaus after ~50 sessions (Supp. Fig. 1**d**). Error bars in **b**-**d** indicate 1 s.d. over 5 runs.

Intuitively, we seek the image $\mathbf{x}$ that most increases the variances along the dimensions in pixel space with small variance $\sigma^2_{\text{small}}$ (i.e., the top eigenvectors of $\Sigma^{-1}$). We do this in order to increase the strength of the weakest signal $\sigma^2_{\text{small}}$ relative to noise $\sigma^2_{\text{noise}}$ (i.e., increase the signal-to-noise ratio $\sigma^2_{\text{small}}/\sigma^2_{\text{noise}}$). Note that the optimization in Eqn. 1 does not depend on previous responses $\mathbf{y}$ nor the current model's weights $\hat{\beta}$. Thus, $\mathbf{x}_{\text{next}}$ can be chosen *before* training the model.

We could choose a new image $\mathbf{x}_{\text{next}}$ from a large image dataset $\mathcal{X}$ based on Eqn. 1. However, this requires taking the inverse of a large $K \times K$ matrix (difficult for computational reasons; $K$ is the number of pixels), and choosing from natural images does not harness the full dynamic range of pixel intensities (i.e., 0 to 255). Instead, we use the intuition of Eqn. 1 to *synthesize* images such that each pixel's variance is maximized. We achieve this by taking a natural image and setting each pixel intensity $p$ to either the maximum value ($p = 255$) or the minimum value ($p = 0$) depending on if $p$ is above or below the mean pixel intensity of the image, respectively. We call the resulting images "gaudy" images for their bright, over-the-top colors (examples in Fig. 2**b**). We confirm that gaudy images increase the variance of each pixel dimension and yield larger objective values in Eqn. 1 than those of normal images (Supp. Fig. 1).

As a first step, we test if gaudy images improve training for a model that satisfies the assumptions of Eqn. 1 (i.e., both model and ground truth mapping are linear). We simulate responses from a Gabor filter model (i.e., an instance of a generalized linear model or GLM). Each response is the result of a linear combination between an input grayscale image and the weights of a Gabor filter, which is then passed through an activation function $f(x)$ (Fig. 1**a**). We then train the filter weights of a GLM with the same activation function but with randomly-initialized filter weights, and we measure performance by predicting responses to heldout normal images (i.e., natural grayscale images, see Methods). We train the GLM over 30 sessions (500 images per session).

We first test the setting in which the Gabor model and the GLM both have linear activation functions (upholding the assumptions of Eqn. 1). As expected, training on gaudy images improves prediction over training on normal images (Fig. 1**b**, orange line above black line). We next consider the setting in which the Gabor model and the GLM both have the same nonlinear activation function. This nonlinearity breaks the linearity assumption of Eqn. 1, but we still find that gaudy images outperform normal images for the relu activation function (Fig. 1**c**) and the sigmoid activation function (Fig. 1**d**). Interestingly, the improved prediction of gaudy images is enhanced as the activation function $f(x)$ increases in nonlinearity (Fig. 1, the differences between the orange and black lines increase from **b** to **d**). This is unexpected, as outliers (i.e., gaudy images) presumably lead to responses at the extremes of the activation functions (e.g., 0 or 1 for the sigmoid) where the derivatives are close to 0 (thus providing no gradient information). However, during the early stages of training, these extremes are rarely encountered. This is because a randomly-initialized GLM reads out a random dimension in pixel space, and it is likely this dimension captures little covariance of the pixel intensities across input images. More important for early training is to choose input images, such as gaudy images,

to drive diverse outputs of this random dimension, which in turn allows error information to more easily back-propagate. Overall, these results suggest that gaudy images may be useful to train DNNs efficiently, as DNNs are built up from layers of GLMs like the ones used here.

## 2 Gaudy images reduce the number of images needed to train DNNs.

Given that gaudy images improve the training data efficiency for GLMs (Fig. 1), we next ask if gaudy images also efficiently trains DNNs, which are essentially feedforward stacks of GLMs. We focus on the real-world, data-limited regression problem faced by visual neuroscientists seeking to characterize visual cortical responses from natural images (Fig. 2**a**). We simulate neural responses from a mid-level visual area (monkey V4) by taking the responses of 100 hidden units (or 'neurons') from a middle layer of a DNN pre-trained for object recognition (Fig. 2**a**, purple, see Methods). The pre-trained DNNs we consider are currently state-of-the-art at predicting V4 responses (explaining ~60% of response variance, Supp. Fig. 6**b**).

Our aim is to predict these 'ground truth' simulated responses from natural images. To do this, we employ transfer learning, a commonly-used approach to predict visual cortical responses [32]. We first pass an image as input into the ResNet50 DNN [33] (trained on ImageNet) and take as features the activity of a middle layer (Fig. 2**a**, blue). We then feed these features as input into a readout network (Fig. 2**a**, green) which in turn outputs a vector of predicted responses (Fig. 2**a**, orange). Importantly, the readout network can capture nonlinear relationships between DNN features and responses and is more expressive than the linear mappings typically employed in current models of higher-order visual cortical neurons [13, 32, 34]. Our goal is to train the readout network (with all pre-trained DNNs held fixed—no fine tuning) to predict simulated responses as accurately as possible with as few training images as possible.

We have designed our simulation setup and training procedure to realistically mimic neurophysiological experiments (see Section 5 for real data results). In a typical experiment, neural activity of ~100 neurons is recorded during a session lasting ~2 hours per day, during which a set of ~500-1,000 unique images are presented to the animal [e.g., 20]. Experiments last for ~30 days. To mimic this setting, we train the readout network for 30 sessions with 500 images per session. We assume the same neurons are recorded across sessions, which is experimentally possible with calcium imaging [35, 36] or neural stitching techniques [37, 38]. However, this need not be the case, and we relax this assumption for real data in Section 5.

Given this setup, we now ask whether training on gaudy images improves the prediction of the readout network more than that achieved by training on colorful, natural images (i.e., "normal" images). We compute a colorful gaudy image by setting each pixel intensity $p$ of a normal image to $p = 0$ if $p$ is less than the mean pixel intensity of the image (taken over all RGB channels), and set $p = 255$ otherwise. The resulting gaudy images have at most eight different colors but still retain a surprising amount of information about the original image (Fig. 2**b**). We find that similar operations (e.g., considering each RGB channel separately or using the median) lead to similar results. In addition, we confirm that gaudy images yield large variances for non-dominant dimensions of the input features for the readout network (Supp. Fig. 2**a**), in line with the optimization goal of Eqn. 1.

We train a readout network with 3 convolutional layers and relu activation functions (see Methods for architecture details of all networks presented in this paper). We measure test performance on heldout normal images (fraction of variance explained averaged over neurons, see Supp. Fig. 2**b** and Methods). We train on either 500 normal images per session or a mix of 250 gaudy images and 250 normal images (where we use a mix instead of 500 gaudy images to avoid too much of a mismatch between training and test set distributions, see Supp. Fig. 2**c**). We find that training on this mix of gaudy images improves prediction, sometimes substantially, versus training solely on normal images (Fig. 2**c**, orange above black lines). This result holds across different pre-trained DNNs used for simulated responses, including VGG19 [39], InceptionV3 [40], and DenseNet169 [41]. When we change all activation functions of the readout network to sigmoid, gaudy images again increase prediction over that of normal images (Fig. 2**d**, orange above black lines). Interestingly, we observe larger improvements in performance for the sigmoid network than for the relu network, similar to the observed boosts in performance for the sigmoid GLM over the relu GLM (Fig. 1**c** and **d**). This suggests that gaudy images are more effective at training DNNs with stronger nonlinear activation functions.

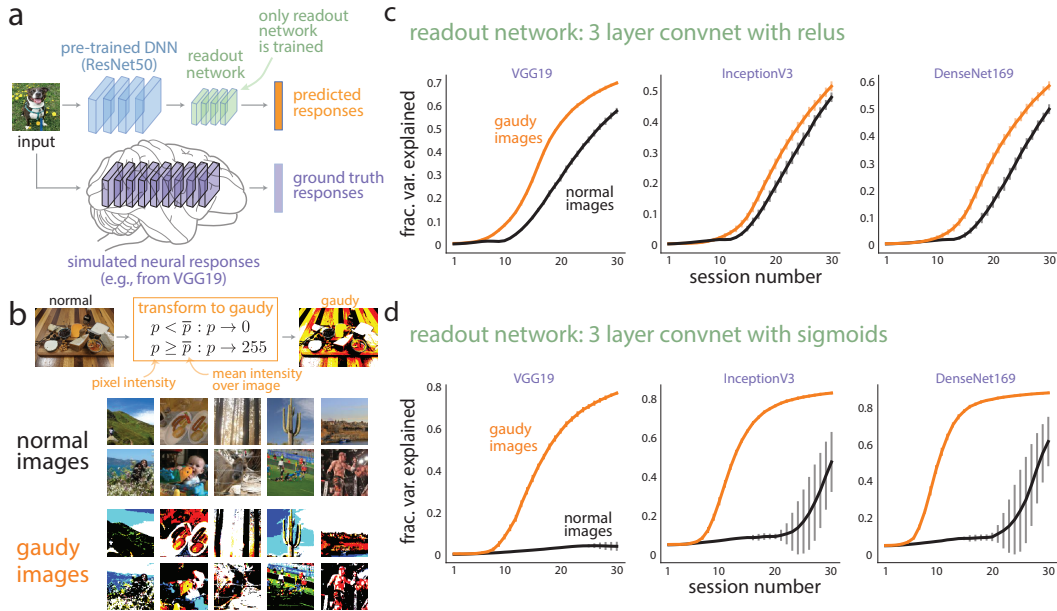

**Figure 2: Gaudy images reduce the number of images needed to train DNNs. a.** Our simulation setup. We simulate responses from higher-order visual cortical areas (e.g., V4 or IT) with responses of hidden units from a middle layer of a DNN trained for object recognition (purple). We predict these "ground truth" responses using a readout network (green) to map the features from a middle layer of a different pre-trained DNN (blue) to predicted responses (orange). Our goal is to train the readout network (all other DNNs are fixed). **b.** Gaudy transformation. Example gaudy images (bottom 2 rows) are brighter and have stronger contrastive edges versus their normal versions (top 2 rows). **c.** Prediction results. We simulate responses from different pre-trained DNNs (each panel). The readout network has 3 convolutional layers with relu activation functions. Performance plateaus after ~50 sessions (Supp. Fig. 3**a**). **d.** Same as **c** except for sigmoid activation functions. Error bars in panels of **c** and **d** indicate 1 s.d. for 5 runs (some error bars are too small to see).

We find that gaudy images improve prediction in additional settings. These settings include readout networks with a large numbers of layers (e.g., 10 layers, Supp. Fig. 2**d**), predicting responses directly from image input without transfer learning (Supp. Fig. 2**e**), and a readout network with an architecture that comprises ResNet-blocks (Supp. Fig. 3**b**). We confirm that these readout networks (i.e., nonlinear mappings) outperform a linear mapping; however, training a linear mapping with gaudy images does not outperform training with normal images (Supp. Fig. 3**c**). This is expected because this setting breaks the assumption of the theory in Eqn. 1 that the ground truth mapping must also be linear. Overall, our results indicate that gaudy images efficiently train DNNs to predict visual cortical responses.

## 3 Gaudy images overemphasize high-contrast edges to improve DNN prediction.

The impressive training data efficiency of gaudy images begs the question: What makes gaudy images so special? From an optimization standpoint, gaudy images likely have two advantages. First, they drive surrogate responses to regions in response space not reachable by normal images. For example, VGG19 responses to gaudy images reside in regions far from those to normal images (Fig. 3**a**, orange dots far from black dots), and responses to gaudy images are more diverse along many response dimensions (Fig. 3**b**, orange line above black line). The diverse responses of gaudy images lead to larger prediction errors (Fig. 3**c**), which in turn lead to larger and more informative gradients. The second advantage is that gaudy images lead to more diverse activations of hidden units in the readout network than those from normal images, even for untrained, random networks (Supp. Fig. 4**a**). This is important because a hidden unit must have its input vary in order to fit the variations of some desired output. Larger variations of this input likely lead to better fits to the variations of the output.

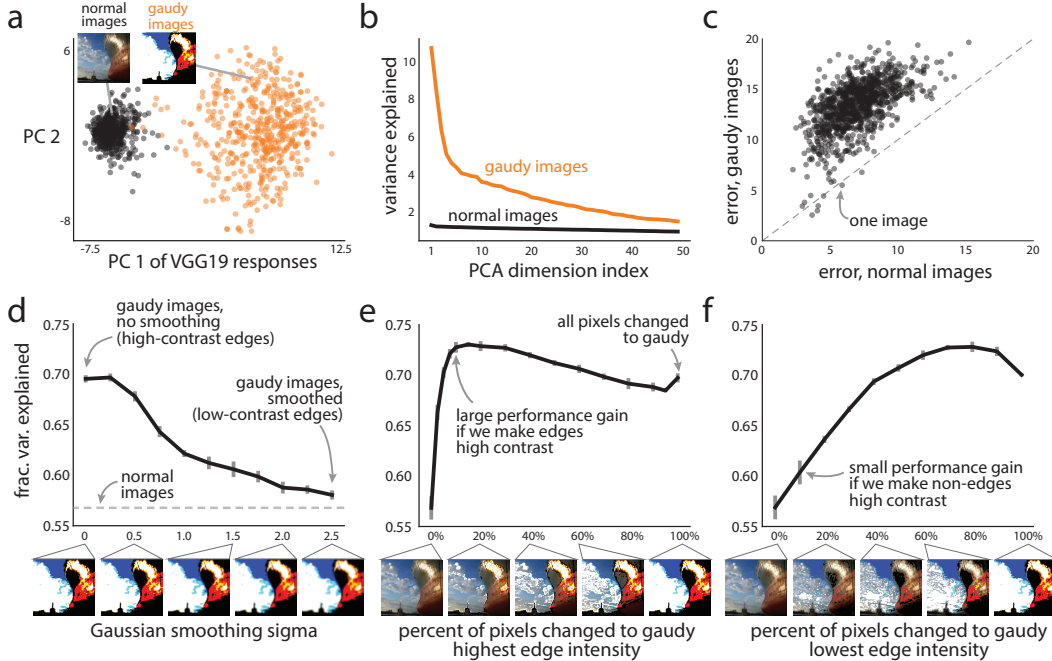

**Figure 3: Gaudy images improve DNN prediction primarily by driving diverse responses and increasing the contrast of edges. a**. The top two principal components (PCs) of VGG19 responses to normal images (black dots) and their gaudy versions (orange dots). **b**. The variance of VGG19 responses for each of the top 50 PCs, where PCA is applied separately to responses from either 5,000 normal images (black) or their gaudy versions (orange). **c**. Error between VGG19 responses and predicted responses of a trained readout network (same as in Fig. 2**c**) to heldout normal images (x-axis) or their gaudy versions (y-axis). **d**. Training on Gaussian-smoothed gaudy images. A Gaussian smoothing sigma of 1.0 corresponds to a s.d. of 1 pixel. **e**. We transform pixels with the highest edge intensities to gaudy (choosing the edge intensity threshold based on percentage quantiles). At 100%, all pixels are transformed to gaudy. **f**. Same as in **e**, except that we transform pixels with the lowest edge intensities to gaudy. In **d**-**f**, we compute the fraction of explained variance ($R^2$ on responses to heldout normal images) using predicted responses from a readout network (same as in Fig. 2**c**) trained after 30 sessions. Error bars in panels **d**-**f** indicate 1 s.d. over 5 runs.

Taken together, gaudy images improve optimization by providing larger prediction errors and more strongly varying the inputs of hidden units.

An important feature of gaudy images is their high contrast—performing the gaudy transformation is akin to substantially increasing an image's contrast (by 400%, Supp. Fig. 4**b**). However, we find a more parsimonious feature in gaudy images that better explains their ability to efficiently train DNNs: Gaudy images overemphasize edges. This intuitively follows from the idea that high-contrast edges strongly drive the edge-detectors of early DNN layers, which in turn more strongly drive feature detectors of later DNN layers. We next show that these high-contrast edges in gaudy images are necessary and sufficient to efficiently train DNNs.

To test for necessity, we smooth the gaudy images (i.e., decreasing the contrast of edges) and find that performance decreases for smoother images (Fig. 3**d**). Thus, high-contrast edges are necessary to increase performance. Next, we ask if high-contrast edges are sufficient to increase performance. To test this, we perform edge detection on each image and compute an edge intensity for each pixel (defined as the norm of the x- and y-gradients using a Sobel edge-detecting filter). Starting with the original image (Fig. 3**e**, 0%), we increase the contrast of edges by transforming a percentage of pixels with the highest edge intensity to gaudy, leaving the remaining pixels unchanged (Fig. 3**e**, compare the clouds and background's silhouette between 0% and 20% image insets). Surprisingly, we find that we need to change only 10% of the pixels with the highest edge intensities to gaudy to achieve a similar (and even larger) performance than that of changing all pixels to gaudy (Fig. 3**e**, compare 10% to 100%). On the other hand, transforming pixels with the lowest edge intensities to gaudy does

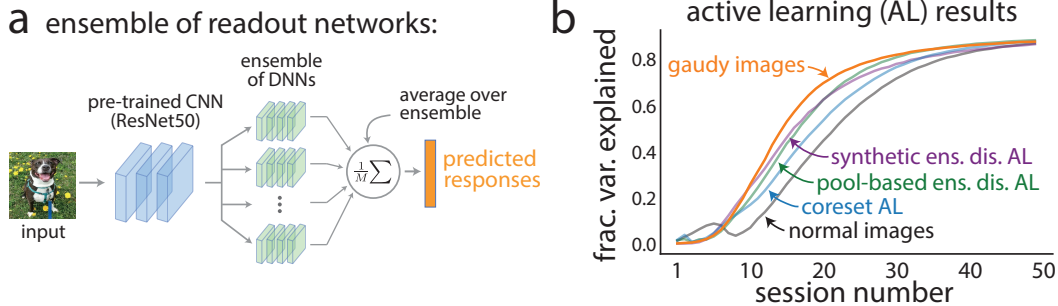

**Figure 4: Gaudy images, chosen *before* training, improve DNN prediction more than that of images chosen *during* training by active learning. a**. Our simulation setup for an ensemble of readout networks (see text for details). **b**. We train the ensemble model with normal images, gaudy images, or images chosen by AL algorithms (described in text). Frac. var. explained ($R^2$) is larger here for an ensemble than for a single DNN (cf. Fig. 2**b**), as expected. The initial "bump" for normal images likely arises from a local optimum that quickly overfits; the bump is present for different learning rates. Example chosen/synthesized images and error bars can be found in Supp. Fig. 5.

not lead to such increases in performance (Fig. 3**f**, 10%). Thus, high contrast edges are sufficient to improve training. Interestingly, transforming 60%-80% of low-edge-intensity pixels still yields an increase in performance above that of 100% (Fig. 3**f**, 60% to 80%), likely because the images for 60%-80% have more high-contrast edges than those for 100% (Fig. 3**f**, 60% vs. 100% image insets). In additional analyses, we find that removing texture (but keeping high-contrast edges) leads to high performance (Supp. Fig. 4**c**), while altering color statistics does not achieve the same level of performance as gaudy images (Supp. Fig. 4**d**). Overall, these results suggest that high-contrast edges are both necessary and sufficient for gaudy images to efficiently train DNNs.

## 4 Gaudy images more efficiently train DNNs than active learning.

We generate gaudy images *before* training. However, it might be the case that adaptively choosing images *during* training (e.g., choosing images based on the model's current uncertainty) will increase training data efficiency even more. Indeed, many adaptive stimulus selection algorithms have been proposed for neuroscientific experiments [17, 19–23, 42–45], but none are equipped to train a DNN with hundreds of thousands of parameters. In machine learning, there has been a recent push to develop active learning (AL) methods to train DNNs for object recognition [46]. These methods include geometric approaches [15], uncertainty approaches [16, 47, 48], adversarial and generative approaches [49–52], among others [53, 54]. However, few studies have proposed AL algorithms for regression problems [55], which require a different notion of uncertainty than that of classification tasks. Thus, to address the regression problem posed in this paper, we propose three AL algorithms, inspired by recent work, to efficiently train DNNs for regression problems. Unexpectedly, we find that gaudy images yield similar and sometimes even larger gains in performance than those for AL algorithms that either access a large number of candidate normal images or synthesize images while preserving natural image statistics.

To test the performance of AL algorithms versus gaudy images, we employ a model for the readout network conducive for AL and then propose three different AL algorithms based on two state-of-the-art AL algorithms for object recognition [15, 16]. The model comprises an ensemble of DNNs, each with the same network architecture but different initial random weights (Fig. 4**a**, green). Each ensemble DNN is trained separately, but the outputs are averaged across the ensemble to compute the predicted responses. Consistent with other ensemble approaches for deep learning [56], we find that an ensemble of DNNs yields better prediction than a single DNN (Supp. Fig. 5**a**). In addition, we find that images with the largest ensemble disagreement (i.e., model uncertainty) also have the largest prediction error (Supp. Fig. 5**b**), suggesting these images will better guide the next gradient step versus randomly-chosen images. The first algorithm ('pool-based ens. dis. AL') is an extension of ensemble approaches for AL [16, 47, 55] and chooses images with the largest ensemble disagreement from a large pool of candidate normal images. The second algorithm ('synthetic ens. dis. AL') synthesizes images to maximize ensemble disagreement using a generator network trained

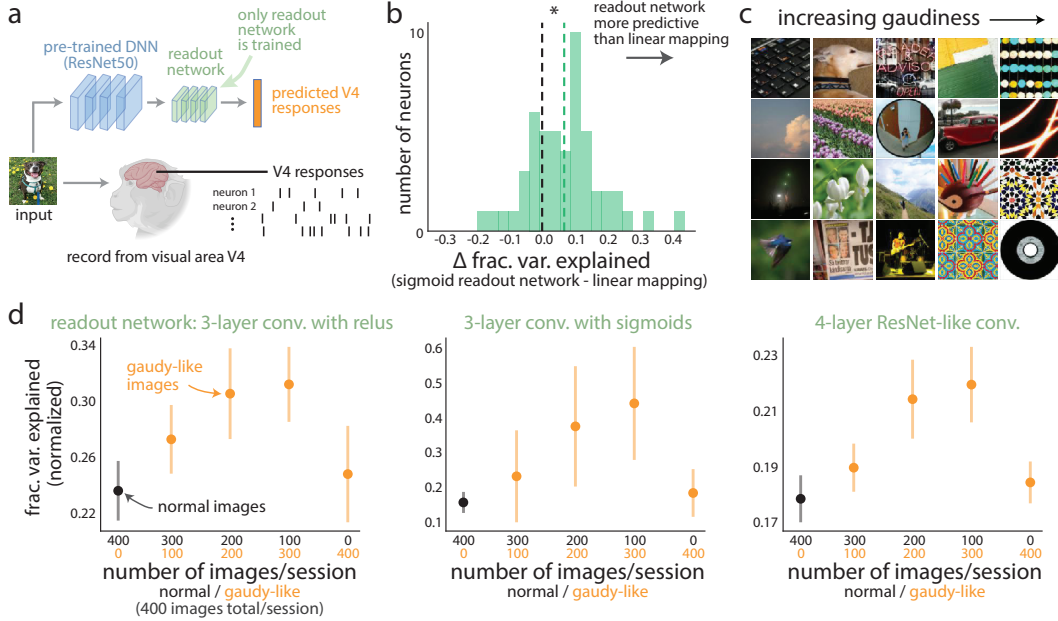

**Figure 5: For real higher-order visual cortical neurons, we find gaudy-like images improve DNN prediction. a**. We train a readout network (green) to predict responses recorded from macaque monkey V4 to natural images. **b**. A sigmoid readout network (trained on 6 sessions) better predicts heldout V4 responses than a linear mapping (average noise-corrected $R^2 = 0.616\%$ vs. $0.548$, $p < 0.002$, permutation test, denoted by asterisk). Fraction variance explained is the $R^2$ between predictions and heldout V4 responses, normalized by an estimated noise-ceiling $R^2$ (see Methods). **c**. Example images of increasing 'gaudiness', defined as the inverse of the MSE between pixel intensities of a normal image and its gaudy version. **d**. We train different readout networks with training images comprised varying proportions of normal and gaudy images (a total of 400 images per session). We train on 6 sessions and compute frac. var. explained on responses from a heldout 7th session. Error bars indicate s.d. over 10 runs with different training session orderings and initial seeds.

in a GAN-like fashion to form a natural prior [57]. The final algorithm ('coreset AL'), extended from previous work [15], uses a coreset approach to choose candidate normal images (from a large pool) that have the largest distance between their corresponding responses and responses to images chosen from previous sessions. Example images chosen by these AL algorithms are presented in Supp. Fig. 5 and further details are in Methods.

We now compare the extent to which gaudy images efficiently train DNNs versus these proposed AL algorithms. Consistent with our findings in the previous sections, gaudy images are more efficient than normal images (Fig. 4**b**, orange line above black line). Moreover, we find that training on gaudy images is more efficient than any AL algorithm (Fig. 4**b**, orange line above purple, green, and blue lines). We find similar results when predicting responses simulated from other pre-trained DNNs (Supp. Fig. 5**c**, performance gains for gaudy images are on par with or larger than those for AL algorithms). These results indicate that gaudy images, chosen *before* training, lead to performances similar to or even greater than those from images chosen adaptively *during* training. This suggests that gaudy images overemphasize features of natural images (e.g., high-contrast edges) that are the most beneficial to efficiently train DNNs in this setting (else the AL algorithms would have identified other features to achieve even better prediction).

## 5 Testing gaudy-like images with real higher-order visual cortical responses.

In the previous sections, we have tested gaudy images with simulations. However, it is unclear if the simulated responses to gaudy images (obtained from pre-trained DNNs) reflect the responses to gaudy images from real visual cortical neurons, suggesting we may not see similar training

improvements when predicting real responses. To address this concern, we consider real neural responses, recorded from macaque monkey V4, to natural images (Fig. 5**a**, see Methods). V4 is a higher-order visual cortical area, and its neurons respond to a wide range of image features, including orientation, spatial frequency, color, shape, texture, and curvature, among others [58–61]. The dataset comprises 7 sessions of ~900 natural images (differing across sessions) and ~40 neurons per session. We confirm that the features of the pre-trained DNNs used in this paper are predictive of these V4 responses (average noise-corrected $R^2$ is ~0.6, consistent with previous studies [13], see Methods). In addition, we confirm that training a readout network better predicts heldout V4 responses than a linear mapping (Fig. 5**b**). This finding supports our claim that a nonlinear mapping (i.e., a readout network) is more predictive than a linear mapping and motivates us to find ways to train readout networks with as few recording sessions as possible.

We next assess if gaudy images are beneficial in training the readout network. Because responses to gaudy images were not present in this dataset, we group the natural images based on their levels of 'gaudiness'. We compute the gaudiness of each image by comparing each image to its gaudy version (Fig. 5**c**). We then consider normal images (i.e., randomly selected images) versus gaudy-like images (i.e., images that are most similar to their gaudy versions). Using the same modeling framework as that for our simulations (Fig. 5**a**), we train each readout network on data from 6 sessions (400 images per session) and test on a heldout 7th session. To account for possibly different recorded neurons across sessions, we allow each session to have different weights in the final dense layer (all other weights are shared across sessions; see Methods). We find that training on only normal images (Fig. 5**d**, black dots) or only gaudy-like images (Fig. 5**d**, rightmost orange dots) achieves worse prediction than training on a mix of normal and gaudy-like images (Fig. 5**d**, orange dots at 200/200 above black dots), consistent with our simulation results (Supp. Fig. 2**c**). In addition to this analysis, we also simulate responses by mapping pre-trained DNN features to V4 responses and find similar benefits for using gaudy images (Supp. Fig. 6). Overall, these results suggest that gaudy images improve DNN prediction of visual cortical responses with fewer recording sessions for training than required by normal images.

## Discussion

We have proposed gaudy images to efficiently train DNNs to predict the responses of visual cortical neurons from features of natural images. In simulations and analyses of real data, we have found that gaudy images increase training data efficiency for all tested DNN architectures and activation functions. We have further found that the high-contrast edges overemphasized in gaudy images are necessary and sufficient for this efficiency increase. Our motivation to use gaudy images comes from a theoretical result in active learning (AL) with linearity assumptions. When we relax those assumptions, we still find that gaudy images, computed *before* training, outperform AL algorithms that choose images *during* training. These results suggest that gaudy images are an important ingredient to efficiently train DNNs.

We have tested gaudy images on somewhat small DNNs (i.e., 1.4 million parameters), and it remains to be seen if gaudy images will be helpful in training larger DNNs (e.g., $> 10$ million parameters) to perform classification tasks (e.g., training ResNet on ImageNet) or unsupervised tasks (e.g., training a generative adversarial network). For data-limited classification tasks, gaudy images may be useful as an image transformation for data augmentation [62–64]. Indeed, on the CIFAR datasets, we have found augmenting with gaudy images increases accuracy (Supp. Fig. 7). We view gaudy images as "pushing the system to its limits" (Fig. 3**a**), and these extreme outputs are helpful in uncovering the underlying computations of the system [18]. An open scientific question is whether gaudy images elicit large responses in visual cortical neurons as they do for DNN hidden units. Going forward, as we learn even more about the structure of natural images relevant to training DNNs, we may better identify the priors that the visual system uses to extract useful information from natural images.

## Broader Impact

The goal of our work is to train DNNs as accurately as possible with as little training data as possible. This reduces the amount of research hours needed to collect data (e.g., an experimenter collecting neural data) and potentially reduces the suffering of the animal. We focus on a specific regression problem of interest in computational neuroscience versus object recognition, for which many active

learning algorithms already exist and would likely require tens of thousands of hours of GPU compute time to perform our analyses. All of our work was performed on a small cluster of eight 12-Gb GPUs (GeForce RTX 2080 Ti). We estimate that ~8,000 GPU hours in total were used, emitting ~200 lbs of $CO2$ (assuming 40 GPU hours consumes 10 kWh). This is equivalent to driving ~230 miles in a car. We do not foresee any short-term negative consequences to society from our work. Code to produce the figures in this paper is available at `https://github.com/pillowlab/gaudy-images`.

## Acknowledgments and Disclosure of Funding

We deeply thank Patricia L. Stan at the University of Pittsburgh and Matthew A. Smith at Carnegie Mellon University for collecting the real neural dataset (Fig. 5). Some elements of figures were created with BioRender.com. B.R.C. was supported by a CV Starr Foundation Fellowship. J.W.P. was supported by grants from the Simons Collaboration on the Global Brain (SCGB AWD543027) and the NIH BRAIN initiative (NS104899 and R01EB026946).

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
