[Supplementary Material]

## Methods

This section describes the image dataset, simulations, network architectures, and neural data used in our work. Code is available at `https://github.com/pillowlab/gaudy-images`. Our simulations were coded in Python, using Keras [65] and TensorFlow [66].

## Image dataset

For our image dataset, we randomly sample 12 million "natural" colorful images from the Yahoo Flickr Creative Commons 100 Million Dataset (YFCC100M) [67], which contains ~100 million images uploaded by users to Flickr between 2004 and 2014. Images are unlabeled (i.e., no content information) and need not contain a recognizable object. We resize each RGB image to $112 \times 112$ pixels, randomly cropping the image to have the same number of row and column pixels. We choose this dataset primarily to ensure that our training images are different from those in ImageNet [68], as the DNNs that we choose for features and simulated responses are trained on ImageNet images.

## Simulations with generalized linear models

We test to what extent gaudy images improve the prediction of generalized linear models (GLMs) when the ground truth model is also a GLM. We grayscale each image ($112 \times 112$ pixels) by averaging over the RGB channels for each pixel. For the ground truth model (Fig. 1**a**), we use a $112 \times 112$ Gabor filter with spatial frequency 0.1 cycles/pixel, bandwidth 0.5, orientation angle 45°, and location at the center of the image. The output response (a single variable) is computed by taking a dot product between the input image and the Gabor filter, which is then passed through an activation function (either linear, relu, or sigmoid). For the sigmoid activation function, we first normalized the dot product (dividing by a constant factor equal to 1,000) before passing it through the sigmoid to ensure responses are within a reasonable range (i.e., not simply 0 or 1). We do not add noise to the ground truth outputs, as we already see training improvements without noise; however, adding output noise leads to similar improvements in prediction when training on gaudy images.

To predict ground truth responses, we consider GLMs with three different activation functions: linear, relu, and sigmoid (Fig. 1**b-d**). The activation function of the GLM always matches that of the ground truth Gabor filter model. This is to ensure that we uphold the assumption made by the active learning theory in Eqn. 1 (i.e., fitting a linear mapping to a ground truth linear function). The GLM takes as input $112 \times 112$ re-centered images (i.e., 110 is subtracted from each pixel intensity). We train the randomly-initialized $112 \times 112$ filter weights of each GLM for 30 "sessions". Each session comprises 500 training images, and GLMs are trained with five passes (or epochs) over the 500 images for that session. To measure test performance, we compute the fraction of variance explained ($R^2$) between ground truth and predicted responses to 4,000 heldout images. For training optimization, we use SGD with momentum, where the learning rates are 1e-7, 3e-7, 1e-7 and the momenta are 0.99, 0.99, and 0.7 for linear, relu, and sigmoid activation functions respectively. Other reasonable learning rates and momentum terms yield similar results. The batch size is 64 images. For each session, we either train on 500 "normal" images (i.e., grayscale images) or 500 "gaudy" images. To compute a gaudy image, we take a normal image and set each pixel intensity $p$ to $p = 0$ if $p < \bar{p}$, else $p = 255$, where $\bar{p}$ is the mean pixel intensity over all pixels of the image.

## Simulations with deep neural networks

We seek to test if gaudy images increasing the training data efficiency of DNNs to predict the responses of higher-order visual cortical neurons to natural images. Ideally, we would predict these responses by training a large DNN with tens of millions of parameters from scratch, end-to-end. However, this would require recording responses to hundreds of thousands of images, currently not possible with the severely limited amount of recording time in neurophysiological experiments. Instead, to model these responses, we employ transfer learning, commonly used by other studies [9, 13, 11, 12]. We pass the input image through a sub-network consisting of the early layers of a DNN pre-trained for object recognition. This output features of this sub-network are then fed as input into a readout network. The readout network in turn outputs the predicted responses. Our goal

is to train the weights of the readout network (keeping the pre-trained DNN fixed) as accurately as possible with as little training data as possible.

To simulate responses of visual cortical neurons, we use the responses of hidden units from middle layers of DNNs (pre-trained on object recognition, different from the pre-trained DNN used to extract features). We find that these pre-trained DNNs predict roughly 40%-60% of the response variance of macaque V4 neurons (Supp. Fig. 6**b**), consistent with previous studies [11, 13]. This suggests that the results of our simulations will likely carry over to neurophysiological experiments.

**Simulating visual cortical responses from pre-trained DNNs**

We consider three different pre-trained DNNs (all trained for object recognition on ImageNet) to simulate responses of higher-order visual cortical neurons. We choose a middle layer of each DNN and perform average pooling ($2 \times 2$ pooling window) to reduce the number of variables. The choice of using a middle layer (versus an early or late layer) is because the image representations of middle DNN layers best match that of visual cortical neurons in monkey V4 [11, 13], our focus in this paper. Here we mention the specific layers we choose; however, choosing other middle layers leads to similar results. The pre-trained DNNs are as follows:

- VGG19 [39]: A popular DNN known for its relatively small convolutional filters ($3 \times 3$). We take the responses from layer `block4_pool`.

- InceptionV3 [40]: A DNN that considers multiple kernels (e.g., $1 \times 1$, $3 \times 3$, and $5 \times 5$) at each stage, where the network's width and depth are balanced. We take the responses from layer `mixed4`.

- DenseNet169 [41]: A DNN where each layer has access to the outputs of all previous layers in an effort to avoid the vanishing gradient problem. We take the responses from layer `pool3_pool`.

To match the number of visual cortical neurons recorded in a typical neurophysiological experiment (~100 neurons), we consider responses from 100 hidden units from the chosen middle DNN layer. Because a randomly-chosen hidden unit likely encodes little stimulus information (as up to 90% of a network's weights can be pruned [69, 70]), we assume that hidden units with the largest response variance carry the most stimulus information. To this end, we apply PCA to the DNN middle layer's responses to 5,000 normal images and take the responses of the top 100 PCs as the simulated responses. Note that each response is a linear combination of the responses of all hidden units from the chosen middle layer (typically ~10,000 hidden units or greater). We scale the weights of each linear combination to ensure the responses of each output neuron have a mean of 0 and a standard deviation of 1.

**Model of higher-order visual cortical neurons: Transfer learning and a readout network**

We employ transfer learning for our model that maps images (i.e., raw pixel intensities) to higher-order visual cortical responses. It consists of a sub-network of a pre-trained DNN (fixed and not trained) followed by a readout network. The sub-network comprises the early layers of ResNet50 [33], which leverages skip connections to increase the number of layers. We perform average pooling ($2 \times 2$ pooling window) for the activations of middle layer `activation_27`, yielding 50,176 output feature variables of shape $7 \times 7 \times 1,024$ (i.e., an activity map of $7 \times 7$ for 1,024 filters). We then consider two readout networks with the same architecture but different activation functions (more network architectures tested in Supp. Fig. 3). Each readout network comprises an initial convolutional layer ($1 \times 1$ kernel, linear activation function, 512 filters), whose purpose is to reduce the number of input variables, followed by a sequence of 3 convolutional layers ($3 \times 3$ kernel, separable convolution, 512 filters) with either a relu activation function (Fig. 2**c**) or sigmoid activation function (Fig. 2**d**). The final layer is a linear, dense layer of 100 filters (to match the number of ground truth DNN neurons, 100). Each readout network has ~1.4 million parameters. We use batch normalization [71] in every layer; without batch normalization, we find it difficult to train DNNs even with a large number of training images (e.g., 50,000 images).

**Training of readout networks**

We initialize each readout network randomly using the default Keras settings, including the random sampling of initial weights from a Glorot uniform distribution. We train networks over 30 sessions, where each session consists of either 500 normal images or a mix of 250 normal images and 250 gaudy images (the latter yielding better performance than training on the full 500 gaudy images per session, Supp. Fig.2**b**). The weights of the trained network from the previous session are carried over to be the initial weights of the network for the current session, and we train the network for 5 passes of the 500 images. We find that re-training a randomly-initialized network each session on data from all previous and current sessions leads to similar results. We optimize with SGD, using a momentum of 0.7 and batch size of 64. The learning rates for the relu readout network are 1e-1, 8e-2, and 7e-2 for ground truth responses of VGG19, InceptionV3, and DenseNet169, respectively. The learning rates for the sigmoid network are 5e-1, 1e0, and 1e0, respectively. In both settings, other reasonable learning rates yield similar results. We test performance by computing the fraction of variance explained ($R^2$) between ground truth and predicted responses to 4,000 normal images, averaged over neurons.

The colorful images are pre-processed and re-sized according to the required RGB format of the respective pre-trained DNN. To compute gaudy images, we set each pixel intensity $p$ to $p = 0$ if $p < \bar{p}$, else $p = 255$, where $\bar{p}$ is the mean pixel intensity over all RGB channels and pixels of the image.

## Simulations with active learning

We ask to what extent gaudy images, chosen *before* training, lead to an increase in DNN prediction versus that achieved by images chosen by active learning (AL) algorithms *during* training. Inspired by the state-of-the-art AL algorithms for deep neural networks performing object recognition [15, 16], we propose three new AL algorithms for deep regression. These AL algorithms choose or synthesize new images that maximize the uncertainty of the model or the diversity of the responses. We use an ensemble of DNNs for our model (Fig. 4**a**) in order to have a measure of uncertainty for AL (i.e., the disagreement among the ensemble DNNs). We randomly initialize each ensemble DNN with a different seed and train each ensemble DNN separately with a different shuffled ordering of training samples. The predicted response of each neuron is the average predicted responses across ensemble DNNs for that neuron. We find that 25 ensemble DNNs yield the best performance (Supp. Fig. 5**a**).

We propose three different AL algorithms. The algorithms use batch learning (i.e., query the responses of chosen images in batches) as opposed to sequentially querying each image. Batch learning is a good fit for neurophysiological experiments, which require time-consuming pre-processing of neural activity (i.e., extracting spike counts from voltage traces or calcium imaging) that is difficult to perform in real-time. For each session, we train with a batch of 500 images, either 500 normal images or a mix of 250 normal images and 250 'non-normal' images (i.e., either gaudy images or images chosen by AL). We use this mix because we find it leads to better DNN prediction (Supp. Fig. 2**b**). The AL algorithms are listed below. We provide example images either chosen or synthesized by these algorithms in Supplemental Figure 5**d**.

- *Pool-based ens. dis. AL* chooses 250 images from a large pool of 80,000 candidate normal images. For each session, a new pool of 80,000 images are randomly chosen from the FlickR image dataset of 12 million images. For each candidate image, we compute a measure of ensemble disagreement (ens. dis.), defined as the median distance between predicted responses of the ensemble DNNs. We find taking the median performs better than computing the variance across ensemble DNNs (summed across neurons), likely because estimates of variance are noisy for a high-dimensional response space (here, 100 dimensions) with few samples (here, responses from 25 ensemble DNNs). Because we choose images in batches, it might be the case that the top 250 candidate images with the highest ensemble disagreement drive similar responses. To account for this possibility, we take the top 2,000 candidate images with the highest ensemble disagreement, and pass these images through the ResNet50 sub-network to obtain their corresponding feature vectors. We then perform a coreset on these feature vectors. The coreset returns 250 of the 2,000 candidate images whose feature vectors are as far as possible from all other feature vectors in feature

space. The images returned by this coreset are the ones used to train the ensemble model for the next session.

- *coreset AL* is an extension of a coreset AL algorithm for DNNs to perform object recognition [15]. For each session, we choose images from a large pool of 80,000 candidate normal images. To begin, we compute the minimum Euclidean distance for each candidate image between the vector of predicted responses to the candidate image and the vectors of predicted responses to all previously-trained images. We then choose the candidate image that has the largest of these Euclidean distances, and append that image to the coreset. We repeat this process 250 times to retrieve 250 chosen images. Our algorithm differs from the previous coreset algorithm [15] by performing coreset on the final output versus the penultimate layer as well training on a mix of 250 normal images and 250 chosen images versus training only on chosen images.

- *Synthetic ens. dis. AL* synthesizes an image based on a natural prior of normal images to maximize the ensemble disagreement. Ideally, synthesizing an image should maximize the distances in response space for all pairs of ensemble DNNs, but this is computationally time-consuming. Instead, we approximately maximize the ensemble disagreement by synthesizing an image in the following procedure. We intialize a candidate image with a randomly-chosen normal image. Then, for each of 20 iterations, we randomly choose two ensemble DNNs, compute the Euclidean distance between the predicted response vectors of the two ensemble DNNs, and use this distance as an objective function for which we can optimize by performing backprop through the network to update the pixel intensities (i.e., the weights of the ensemble DNNs remain fixed). However, because updating pixel intensities directly leads to adversarial examples [72], we further backprop through a generator network that maps a feature vector to an image. The idea is to synthesize images by taking a gradient step along the manifold of natural images (i.e., feature space) rather than a gradient step in pixel space. We use a similar network architecture and GAN training procedure from a previous study for the generator and discriminator networks [57]. As feature vectors, we use the output features of the ResNet50 sub-network, and we use the same sub-network for the comparator network. We train the generator and discriminator networks on our 12 million image dataset. Because we perform batch AL, we also incorporate a diversity term in the objective function equal to the minimum Euclidean distance between the feature vector of the current synthesized image and all previously-converged synthesized feature vectors for that session. This term encourages diversity by ensuring that the current synthesized feature vector is far from any previously-synthesized feature vectors. To recover the 250 synthesized images, we pass these feature vectors through the generator network.

## Real neural data results

To confirm that our simulation testbed in Figure 2 is realistic, we perform further tests with real neural data (visual cortical responses recorded from macaque visual area V4).

### Experiment details

All neural data was collected by Patricia L. Stan and Matthew A. Smith at Carnegie Mellon University. Here, we briefly describe the neural data collection; see previous studies for details of almost-identical experiments [20, 73]. A macaque monkey was implanted with a 96-electrode array in visual area V4. On each trial, the awake animal fixated on a central dot while 6 images flashed in the aggregate receptive fields of the recorded V4 neurons. Colorful natural images (example images in Fig. 5c) were presented for 100 ms interleaved with 100 ms isoluminant blank screens. After the sixth flash, the animal made a saccade to a target dot (whose location was unrelated to the shown images) and received a water reward. There were 7 recording sessions in total; 6 of these sessions had 900 presented images and one session had 1,083 presented images. Each image was repeated at least 10 times. All images were shown once in a random sequence before showing the same images again in a different random sequence. Voltages recorded from each electrode were spike-sorted, and each session yielded responses to ~50 V4 neurons. Each response was the repeat-average of spike counts for the 100 ms presentations of the same image (shifted to be taken 50 ms after stimulus onset). The resulting data from each session is the set of shown images and a response matrix where each row corresponds to one V4 neuron and each column corresponds to one shown image.

**Identifying gaudy-like images**

Because gaudy images were not presented to the animal in these recording sessions, we do not know how V4 neurons respond to gaudy images. However, we can identify the most gaudy-like images as proxies for true gaudy images. To do this, we compute the level of "gaudiness" for each image by taking the mean-squared-error (MSE) between the RGB pixel intensities of the original image and its gaudy version. Images with a small MSE are most gaudy-like (see Fig. **5c** for example images with varying levels of gaudiness).

**Predicting visual cortical responses with a linear mapping**

Visual cortical responses are predicted via a linear mapping either from features from a pre-trained DNN (e.g., VGG19) or from the output embeddings of a readout network. To estimate the linear mapping, we use ridge regression with regularization parameter $\lambda$ equal to 1% of the total summed variance of the input features used for training. We use 4-fold cross-validation.

**Normalized fraction of variance explained**

To compute performance, we compute a normalized fraction of variance explained, following a similar procedure as in previous studies [11, 22]. In brief, this fraction is the $R^2$ between predicted responses and V4 responses, divided by an $R^2$ that represents the maximum $R^2$ achievable under noise across repeats. For each session, we divide an image's repeats into two equal-sized splits (typically ~5 repeats) and compute the repeat-averaged responses for each split. We then compute the maximum $R^2$ by taking the $R^2$ between the repeat-averaged responses between the two splits across all images. To compute the $R^2$ between predicted responses and V4 responses, we choose one of the two splits of responses and compute cross-validated predictions for all images. We then compute the $R^2$ between the predicted responses and the heldout responses (from the same split of responses as those used to train the linear mapping) to images across all folds. Note that this approach differs slightly from previous studies in that we compute an $R^2$ across all images instead of taking an $R^2$ for each cross-validation fold and returning the $R^2$ averaged over folds. This performance metric is different than that reported in BrainScore [13], which uses correlation instead of $R^2$.

**Predicting V4 responses with a readout network**

We considered the same readout network architectures as in Fig. 2 (relu and sigmoid networks) and Supp. Fig. 3 (ResNet-like network). The output of the second-to-last layer of each readout network is a set of 512 embedding variables (akin to how the features of the pre-trained DNNs are outputs of a middle layer). Each V4 neuron is a linear readout of these embedding variables.

Each readout network is trained as follows. Out of the 7 recording sessions, we choose 6 sessions for training (each having 900 natural images) and hold out the last session for testing (1,083 natural images). One epoch of training consists of a pass through each of the 6 training sessions; we train for 10 epochs. For each epoch, the session ordering is randomly shuffled. Because it is not certain each session has the same recorded V4 neurons (i.e., the number of V4 neurons is not the same across sessions), we randomly re-initialize the weights of the last layer (i.e., the linear readout of the embedding variables) and adjust the number of output variables to match the number of V4 neurons for that session. Due to this random re-initialization, we train on each session 5 times before moving on to another session during an epoch. We use learning rates 5e-2, 1e-1, and 5e-2 for the relu, sigmoid, and ResNet-like readout networks, respectively. All other optimization hyperparameters are kept the same as in Fig. 2. For testing, we fit a linear mapping from the embedding variables to heldout V4 responses from a separate session (see above section 'Normalized fraction of variance explained.'). Thus, we test on V4 neurons and natural images unobserved by the readout network.

# Appendix

Here we formally derive the active learning (AL) theory in Equation 1. This theory presents the optimal strategy for training a linear regression with AL. A key assumption is that the ground truth mapping is also linear. A counterintuitive result from this theory is that the optimal stimuli can be chosen *before* collecting training data as opposed to typical AL algorithms that optimize stimuli *during* the data collection and training process. We have adapted this theory from previous work [30, 31] and present it in the context of predicting neural responses from images.

Let us assume a linear mapping between image $\mathbf{x} \in \mathbb{R}^K$ (for $K$ pixels) and neural response $y$: $y = \beta^T \mathbf{x} + \epsilon$, with weight vector $\beta \in \mathbb{R}^K$ and noise $\epsilon \sim \mathcal{N}(0, \sigma_{\text{noise}}^2)$. We model this mapping with $\hat{y} = \hat{\beta}^T \mathbf{x}$, where $\hat{\beta} = \Sigma^{-1} \mathbf{X} \mathbf{y}$ for a set of (re-centered) training images $\mathbf{X} \in \mathcal{R}^{K \times N}$ (for $K$ pixels and $N$ images), unnormalized covariance matrix $\Sigma = \mathbf{X} \mathbf{X}^T$, and responses $\mathbf{y} \in \mathbb{R}^N$. We can compute the expected error that represents how well our estimate $\hat{\beta}$ matches that of ground truth $\beta$:

$$
\begin{aligned}
\mathrm{E}\Big[\|\beta - \hat{\beta}\|_2^2 \mid \mathbf{X}\Big] &= \mathrm{E}\Big[\|\beta - \Sigma^{-1}\mathbf{X}\mathbf{y}\|_2^2\Big] = \mathrm{E}\Big[\|\beta - \Sigma^{-1}\mathbf{X}\big(\mathbf{X}^T\beta + \underline{\epsilon}\big)\|_2^2\Big] = \\
&\mathrm{E}\Big[\|\Sigma^{-1}\mathbf{X}\underline{\epsilon}\|_2^2\Big] = \mathrm{E}\Big[\mathrm{Tr}\big[(\Sigma^{-1}\mathbf{X}\underline{\epsilon})^T(\Sigma^{-1}\mathbf{X}\underline{\epsilon})\big]\Big] = \sigma_{\text{noise}}^2\, \mathrm{Tr}(\Sigma^{-1})
\end{aligned}
\tag{2}
$$

where $\underline{\epsilon} \in \mathbb{R}^N$ is the vector of noise values for each response. We seek a new, unshown image $\mathbf{x}_{\text{next}}$, from the set of all possible (re-centered) images $\mathcal{X}$, that minimizes the error in Eqn. 2 when we train $\hat{\beta}$ on the new image $\mathbf{x}_{\text{next}}$ and its response $y_{\text{next}}$. Importantly, we do not know $y_{\text{next}}$, so we cannot ask which image has the largest prediction error $\|y_{\text{next}} - \hat{y}(\mathbf{x}_{\text{next}})\|_2^2$. Instead, we choose the image $\mathbf{x}_{\text{next}}$ that minimizes the error between the true $\beta$ and our estimate $\hat{\beta}$ trained on images $[\mathbf{X}, \mathbf{x}_{\text{next}}] \in \mathbb{R}^{K \times (N+1)}$:

$$
\begin{aligned}
\mathbf{x}_{\text{next}} = \min_{\mathbf{x} \in \mathcal{X}} \mathrm{E}\Big[\|\beta - \hat{\beta}\|_2^2 \mid [\mathbf{X}, \mathbf{x}]\Big] &= \min_{\mathbf{x}} \sigma_\epsilon^2\, \mathrm{Tr}\big[(\Sigma + \mathbf{x}\mathbf{x}^T)^{-1}\big] \overset{(\triangle)}{=} \\
\min_{\mathbf{x}} \mathrm{Tr}(\Sigma^{-1}) - \frac{\mathbf{x}^T \Sigma^{-1}\Sigma^{-1}\mathbf{x}}{1 + \mathbf{x}^T \Sigma^{-1}\mathbf{x}} &= \max_{\mathbf{x}} \frac{\mathbf{x}^T \Sigma^{-1}\Sigma^{-1}\mathbf{x}}{1 + \mathbf{x}^T \Sigma^{-1}\mathbf{x}}
\end{aligned}
\tag{3}
$$

where in $(\triangle)$ we apply the Sherman-Morrison formula. The objective function of the rightmost formula in Eqn. 3 reveals that the image $\mathbf{x}$ that decreases the error the most is the one that maximizes the magnitude of its projection along the eigenvectors of $\Sigma^{-1}$ with the largest eigenvalues. An easier way to intuit this optimization is to assume that the $K$ pixels of $\mathbf{X}$ are uncorrelated (e.g., via a change of basis). Under this setting, the covariance matrix of pixel intensities $\Sigma$ is a diagonal matrix with entries $\Sigma_{k,k} = \sigma_k^2$. The optimization becomes the following:

$$
\mathbf{x}_{\text{next}} = \arg\max_{\mathbf{x} \in \mathcal{X}} \frac{\mathbf{x}^T (\Sigma^{-1})^2 \mathbf{x}}{1 + \mathbf{x}^T \Sigma^{-1} \mathbf{x}} \overset{\Sigma = \mathrm{diag}(\Sigma)}{=} \arg\max_{\mathbf{x}} \sum_{k=1}^{K} \mathbf{x}_k^2 / \sigma_k^2
\tag{4}
$$

where $\Sigma = \mathrm{diag}(\Sigma)$ indicates that the intensities of each pair of pixels is uncorrelated. Intuitively, we seek the image $\mathbf{x}$ that maximizes the magnitude of its projection along dimensions in pixel space with small variance $\sigma_{\text{small}}^2$. We do this in order to increase the strength of the weakest signal $\sigma_{\text{small}}^2$ relative to $\sigma_{\text{noise}}^2$ (i.e., increase the signal-to-noise ratio $\sigma_{\text{small}}^2 / \sigma_{\text{noise}}^2$). Note that the optimization in Eqn. 3 does not depend on previous responses $\mathbf{y}$ nor the current model's weights $\hat{\beta}$. Thus, $\mathbf{x}_{\text{next}}$ can be chosen *before* training the model.

**Supplementary Figure 1: Assessing the extent to which gaudy images increase the variance of pixel dimensions and whether performance plateaus after training on gaudy images for many sessions.**

**a**. We apply PCA to 10,000 grayscale images ($112 \times 112$ pixels) to compute the eigenvectors of the top 2,000 PCs. We then compute the variance of each PC dimension for different 10,000 grayscale images and their gaudy versions. Gaudy images (orange line) yield larger variances than those of normal images (black line). PCA is applied separately to either normal or gaudy images.

**b**. Ratios of variance explained (variance of gaudy images divided by variance of normal images) for each PC index. Gaudy images yield larger variances for pixel dimensions that explain a small amount of variance than those of normal images (i.e., ratios increase with PC index). This suggests that gaudy images satisfy the optimal strategy derived from the AL theory in Eqn. 1: Maximize the variance for low-variance pixel dimensions.

**c**. We further confirm that gaudy images yield larger objective values of Eqn. 1 than those of normal images (dots are above gray dashed line, $p < 0.002$, permutation test). In the objective function, $x$ is a (re-centered) image with $K$ pixels, and $\Sigma$ is the $K \times K$ covariance matrix of pixel intensities.

**d**. Same as in Fig. 1**b-d** except that we train for 100 sessions instead of 30 sessions. For the linear (left panel) and relu (middle panel) activation functions, performance plateaus after ~50 sessions with no appreciable difference in the final performance between training either on normal or gaudy images. For the sigmoid activation function (right panel), more sessions are needed to reach a plateau. Error bars indicate 1 s.d. for 5 runs (some error bars are too small to see).

**Supplementary Figure 2: Additional results for using gaudy images to train DNNs: Assessing the residual error of predictions, varying the number of gaudy images trained per session, varying the number of DNN layers, and testing gaudy images with no transfer learning.**

(continued on next page...)

**Supplementary Figure 2:** (...continued from previous page.)

**a**. For grayscale images, we find that gaudy images increase the variance of low-variance pixel dimensions (Supp. Fig. 1**a** and **b**). Here, we find that gaudy images increase the variance of low-variance dimensions in the space of ResNet50 features. To see this, we apply PCA (taking the top 2,000 PCs) to the ResNet50 features of 10,000 normal images to identify the PCA loadings. We then compute the variance of these PC dimensions for ResNet50 features to a different set of 10,000 normal images and their gaudy versions. We find that the ratio of the variance of gaudy images and that of normal images is above 1 for the low-variance PC dimensions (i.e., PC dimensions 1,000-2,000). This suggests that gaudy images make more prominent these low-variance dimensions for the readout network, a desired property derived from the AL theory in Eqn. 1.

**b**. Predicted responses versus ground truth responses for an example simulated neuron from VGG19 (same neuron in both panels). Training on gaudy images (right panel) leads to smaller errors between predicted and ground truth responses than those for training on normal images (left panel). This is especially true for outliers (i.e., ground truth responses with large magnitudes). Predicted responses are from training a 3 layer convolutional network with relu activation functions (same readout network as in Fig. 2**c**) after 30 sessions. Each dot corresponds to one heldout normal image. Fraction of variance explained ('frac. var. explained') is computed from responses to 4,000 heldout normal images. This example neuron has the median frac. var. explained of all VGG19 surrogate neurons when trained on normal images.

**c**. For each session, we train a DNN with 500 images. One design choice is the number of gaudy images to use per session (the rest being normal images). We test this by training (for 30 sessions) a 3 layer convolutional network either with relu activation functions (left panel) or with sigmoid activation functions (right panel) to predict VGG19 responses to heldout normal images. We find that the setting in which all 500 training images are gaudy images does not achieve the best prediction (rightmost point in each panel). Instead, a mix between gaudy and normal images achieves the best prediction (e.g., 250 normal and 250 gaudy images). The reason this mix performs best is likely because responses to gaudy images tend to be outliers (Fig. 3**a**), and training on only gaudy images leads to a large mismatch between the training and test set distributions. Instead, a 50%-50% mix between gaudy and normal images represents a trade-off between training on samples with large errors (i.e., gaudy images) versus ensuring that the training set distribution matches the test set distribution (i.e., normal images). Thus, this mix is likely an important ingredient to train a DNN with data augmentation or AL methods. We use this 50%50% mix to train readout networks to predict simulated responses in all other analyses in this paper.

**d**. We vary the number of layers of the readout network (i.e., a $K$-layer convolutional network with 512 filters per layer, similar to readout network in **c**) from 1 to 9 layers. We find that training on gaudy images outperforms training on normal images for all numbers of layers (orange line above black line). This includes networks with relu (left panel) and sigmoid (right panel) activation functions. Networks are trained to predict VGG19 responses. We use a learning rate of 1e-1 for all numbers of layers for the relu networks (trained for 20 sessions) and a learning rate of 5e-1 for the sigmoid networks. We find that the deeper sigmoid networks (e.g., 6 or more layers) require more training data, so we increase the number of sessions to equal the number of layers multiplied by 10 (e.g., for 10 layers, we train for 100 sessions), except for 1 layer (for which we trained for 20 sessions).

**e**. Our main simulation testbed (Fig. 2**a**) relies on transfer learning to map an input image to predicted responses. Our setup makes the most sense for this data-limited regime, where we can utilize prior observations that the features from a middle layer of a pre-trained DNN are predictive of visual cortical responses. However, ultimately one would desire to train an end-to-end DNN without the (possibly incorrect) priors of transfer learning. Thus, we test if gaudy images are also helpful in training DNNs end-to-end (left panel)—inputting images directly into the readout network instead of inputting ResNet50 features (compare **e**, left panel, with Fig. 2**a**). We use the same architectures for the readout networks with relu and sigmoid activation functions as those in Fig. 2**c** and **d**, except that we add an average pooling layer (with $2 \times 2$ kernel) after each convolutional layer in order to reduce the spatial size of the output of these layers.

(continued on next page...)

**Supplementary Figure 2:** (...continued from previous page.)

We perform training and optimization (learning rates: 5e-2 and 5e-1 for the relu and sigmoid networks, respectively) in a similar manner as in Fig. 2. One key difference is that instead of keeping the ratio of gaudy images and normal images fixed (i.e., 250 gaudy/250 normal for a total of 500 training images per session), we linearly decreased this ratio from 250 gaudy/250 normal for the first session to 100 gaudy/400 normal for the 30th session. We find that dynamically changing this ratio leads to better results. Overall, we find that gaudy images lead to an increase in end-to-end DNN prediction with less training data needed than that of normal images. We note, however, that DNN prediction without transfer learning is less than that with transfer learning (compare **d** and **e**), suggesting transfer learning is appropriate for our task.

For panels **c**, **d**, and **e**, error bars indicate 1 s.d. for 5 runs (some error bars are too small to see).

**a**  readout network: 3-layer convnet with relus (100 sessions)

simulated response DNN: VGG19     InceptionV3     DenseNet169

gaudy images

normal images

performance drops due to train and test set mismatch

gaudy images improve training

realistic # sessions for biological experiments

frac. var. explained

session number

**b**  readout network: 4-layer convnet with ResNet-block structure

VGG19     InceptionV3     DenseNet169

gaudy images

normal images

frac. var. explained

session number

**c**  readout network: linear mapping (no hidden units)

VGG19     InceptionV3     DenseNet169

gaudy images

normal images

frac. var. explained

session number

**Supplementary Figure 3: Additional results for using gaudy images to train DNNs: Training a DNN for many sessions, training a DNN with a ResNet-like architecture, and training a linear mapping.**

(continued on next page...)

**Supplementary Figure 3:** (...continued from previous page.)

**a**. Results for training the same readout network as in Fig. 2**c** (i.e., 3 layer convolutional network with relu activation functions) for 100 sessions. Performances for the first 30 sessions are the same as those in Fig. 2**c**. However, after ~50 sessions, performance plateaus, and we observe a slight increase in performance for training on normal images versus training on gaudy images (black line slightly above orange line after 50 sessions). This is expected, as gaudy images represent a mismatch between training and test set distributions, whereas no mismatch exists when training on a large number of normal images. Thus, adopting a hybrid approach of initially training on gaudy images and then transitioning to training on normal images will likely yield the best results. For training DNNs to predict visual cortical responses, we suspect the number of training images needed to begin to transition to training only on normal images (here, 25,000 images) is likely out of reach for most neurophysiological experiments.

**b**. Results for a readout network with 4 ResNet blocks [33]. We find that gaudy images (orange) substantially improve prediction over normal images (black) for this network architecture. Each ResNet block consists of 3 convolutional layers, where the first two layers have 256 filters ($3 \times 3$ kernel, stride of 1, separable convolution) and the last layer has 512 filters ($1 \times 1$ kernel, stride of 1, separable convolution). We include a skip connection for the last operation of each block, which sums the output of the block's last layer to the block's input. The first layer of the network is a convolutional layer with 512 filters ($1 \times 1$ kernel, linear activation function), whose output is then passed to the first ResNet block. The final layer is a linear mapping between the output of the last ResNet block and the predicted responses. We initialize the network randomly with default Keras settings, including the sampling of initial weights from the Glorot uniform distribution. We use a learning rate of 2e-1 and a momentum value of 0.7. The testing procedure is the same as that for Figure 2**c** and **d**.

**c**. For completeness, we also train a linear mapping for the readout network. Here, because the true mapping between features and responses is likely nonlinear, we do *not* expect gaudy images to improve training. This is because the AL theory for linear regression (Eqn. 1) assumes that the ground truth mapping between features and responses is linear—which is not the case here. We fit the linear mapping using SGD with a learning rate of 1e-2 and a momentum value of 0.7.

In our work, we use a nonlinear mapping (i.e., the readout network in **a**) between the features of a pre-trained DNN and responses. However, a linear mapping may perform as well as or better than a nonlinear mapping. Indeed, for a small amount of training data, we find this to be the case (compare performance for VGG19, session 10, between **a** and **c**, left panels). However, when we train both mappings for 100 sessions, the performance of the readout network (fraction variance explained of $0.873 \pm 0.002$) is larger than that of a linear mapping ($0.595 \pm 0.001$, where $\pm$ indicates 1 s.d.). Thus, in this setting, a nonlinear mapping, trained on enough data, is more appropriate than a linear mapping when mapping DNN features to responses.

For all panels, error bars indicate 1 s.d. for 5 runs (some error bars are too small to see).

**a**

random, untrained relu network

random, untrained sigmoid network

**b**

400% increase in contrast → reaches performance level of gaudy images

gaudy images

normal images

percent increase in contrast

**c**

gaudy images

normal images

Gaussian smoothing sigma

**d**

gaudy images

normal images

grayscale / black and white only / red only / blue only / green only / inverted / above mean only / below mean only / gaudy

different transformations to RGB color channels

**Supplementary Figure 4: Additional results to understand which features of gaudy images are the most important for improving training.**

(continued on next page...)

**Supplementary Figure 4:** (...continued from previous page.)

**a**. To understand how gaudy images affect the optimization of DNN weights, we consider to what extent gaudy images drive diverse activity in the readout network. Diverse activity is advantageous for optimization, as the input variables of a hidden unit need to vary in order to regress to the desired output variable.

Here, we compute the variance of responses to either 1,000 normal or 1,000 gaudy images in randomly-initialized, untrained readout networks (same architectures as in Fig. 2**c** and **d**). We find that gaudy images yield significantly larger output variances for hidden units in the second layer, both for a randomly-initialized relu network (left panel, $p < 0.001$, permutation test) and a randomly-initialized sigmoid network (right panel, $p < 0.001$, permutation test). This suggests that gaudy images lead to more variation in the input variables of each hidden unit (especially for the small number of images in a batch), which in turn makes it easier to identify the most informative input variables for that hidden neuron.

**b**. The gaudy transformation is akin to substantially increasing the contrast of a normal image. Here, we train the readout network on images with varying levels of contrast to infer the extent to which the gaudy transformation increases contrast. We change contrast with the commonly-used contrast stretching approach [74]. Consider a percent contrast increase $c$ (e.g., $c = 10$ indicates a 10% increase in contrast). For each image, we compute a minimum luminance $m$ as the 5% quantile of pixel intensities for that image. Then, for each pixel intensity $p$, we compute a new pixel intensity $\tilde{p} = (p - m)(100 + c)/100$, and then clip $\tilde{p}$ to be between 0 and 255. The bottom inset is a set of example images for different percent increases in contrast. The training and testing procedure, as well as the readout network, are the same as in Figure 3**d**, **e**, and **f**. We find that gaudy images represent an increase of 400% in contrast, as 400% achieves similar performance as that of gaudy images. This increase is substantially larger than that used in data augmentation methods, typically no more than 50% [75].

**c**. In Figure 3**d**, we transform an image to a gaudy image and then perform Gaussian smoothing, eliminating any high-contrast edges. Here, we perform this procedure in reverse (i.e., first smooth a normal image and then transform to gaudy) to further test if the high-contrast edges of gaudy images are important features to improve training. A Gaussian smoothing sigma of 1.0 corresponds to a s.d. of 1 pixel. Under this procedure, the smoothed gaudy images still retain high-contrast edges but lose high-frequency spatial information such as textures (inset, rightmost example image). Even for highly-smoothed gaudy images, performance is similar to that of gaudy images (cf. sigma values of 0 and 10). This result provides further evidence that high-contrast edges are likely the most important features of gaudy images to improve training.

**Supplementary Figure 4:** (...continued from previous page.)

**d**. We also test if different color transformations lead to different training performances (bottom inset is a set of example images, one for each transformation). We find that these color transformations do improve prediction above that of normal images (gray dashed line) but not above that of gaudy images (orange dashed line and rightmost dot). The image transformations are as follows (with $\bar{p}$ as the mean pixel intensity of the image):

- **grayscale**: Transform each pixel intensity to its across-channel mean intensity, then transform to gaudy (see **gaudy** below).

- **black and white only**: Transform to gaudy only pixels whose intensities across channels are either all above $\bar{p}$ or all below $\bar{p}$. Pixels that do not satisfy these criteria are not transformed and remain normal.

- **red only**: Transform to gaudy the pixel intensities of the red-channel only (green and blue channel intensities remain normal) with threshold $\bar{p}$.

- **green only**: Same transformation as **red only** except only for pixel intensities of the green-channel.

- **blue only**: Same transformation as **red only** except only for pixel intensities of the blue-channel.

- **inverted**: An inversion of the gaudy transformation. If pixel intensity $p < \bar{p}$, the gaudy pixel intensity $\tilde{p} = 255$, else $\tilde{p} = 0$.

- **above mean only**: If pixel intensity $p \geq \bar{p}$, then the gaudy pixel intensity $\tilde{p} = 255$, else $\tilde{p} = p$. The resulting image emphasizes "bright" regions of the image.

- **below mean only**: Opposite of **above mean only**. If $p < \bar{p}$, then $\tilde{p} = 0$, else $\tilde{p} = p$. The resulting image emphasizes "dark" regions of the image.

- **gaudy**: The gaudy transformation. If pixel intensity $p < \bar{p}$, then we set the gaudy pixel intensity $\tilde{p}$ to $\tilde{p} = 0$, else $\tilde{p} = 255$.

For panels **b**, **c**, and **d**, error bars indicate 1 s.d. for 5 runs (some error bars are too small to see).

a

fraction var. explained

more ensemble DNNs
↳ better performance

number of ensemble DNNs

b

error (heldout)

one image

$\rho = 0.90$

ensemble disagreement

c

VGG19

frac. var. explained

normal images | pool-based ens. dis. AL | coreset AL | synthetic ens. dis. AL | gaudy images

InceptionV3

normal images | pool-based ens. dis. AL | coreset AL | synthetic ens. dis. AL | gaudy images

DenseNet169

normal images | pool-based ens. dis. AL | coreset AL | synthetic ens. dis. AL | gaudy images

d  example images chosen during session 20

normal images | pool-based ens. dis. AL | coreset AL | synthetic ens. dis. AL | gaudy images

**Supplementary Figure 5: Gaudy images, chosen *before* training, improve the training of DNNs on par with or greater than active learning (AL) algorithms, which choose images *during* training.**

(continued on next page...)

**Supplementary Figure 5:** (...continued from previous page.)

**a**. For the AL experiments, we use an ensemble of readout DNNs (Fig. 4**a**), primarily because we can then use the disagreement among the ensemble DNNs as a measure of uncertainty for AL. Here, we vary the number of ensemble DNNs and train them (over 20 sessions with normal images, where each ensemble DNN is a 3 layer convolutional network with relu activation functions) to predict ground truth VGG19 responses. We find that increasing the number of ensemble DNNs improves performance (compare 1 to 25 ensemble DNNs). We use 25 ensemble DNNs in all other analyses in Figure 4 and this figure.

**b**. The ensemble disagreement (defined as the median distance of predicted response vectors among ensemble DNNs, see Methods) is significantly correlated to the error of heldout images (Pearson's correlation $\rho = 0.90$, $p < 0.001$, permutation test). Thus, choosing images with high ensemble disagreement (for which we have access during training) is akin to choosing images with large prediction error (for which we do *not* have access during training). Images with large prediction error yield larger, more informative gradient steps which in turn train the networks with less training data.

**c**. Prediction results for a large range of AL algorithms across different pre-trained DNNs used to simulate visual cortical responses. The AL algorithms (purple) are the same as in Fig. 4**b**. Models are trained in the same manner as in Figure 4**b** for 20 sessions. We find that gaudy images either outperform or are on par with the tested AL algorithms (orange bars above or equal to other bars). This suggests that gaudy images overemphasize natural image features (e.g., high-contrast edges) that aid in training, and that these features are the most important for training (else AL algorithms would have identified other features to increase their performance over that of gaudy images).

Error bars in **a** and **c** indicate 1 s.d. over 5 runs (some error bars are too small to see).

**d**. Example images chosen or synthesized by the different AL algorithms on session 20 when predicting VGG19 responses. The pool-based ens. dis. AL and coreset AL algorithms tend to find bright, textured images, while the synthetic ens. dis. AL algorithm appears to synthesize gaudy-like images.

**Supplementary Figure 6: Simulation testbed where simulated responses come from a linear mapping from pre-trained DNN features to visual cortical responses from macaque V4.**

(continued on next page...)

**Supplementary Figure 6:** (...continued from previous page)

**a**. Simulation setup. In Figure 2, simulated responses are linear combinations of the features from a pre-trained DNN (e.g., VGG19). One concern is that these simulated responses may not reflect true visual cortical responses. To address this, we consider simulated responses that leverage real visual cortical responses. The neural dataset comprises 7 recording sessions of neural responses recorded from macaque visual area V4 to presentations of natural images (see Methods). For each session, we fit a linear mapping from the features of a pre-trained DNN to the V4 responses. We then append this linear mapping as the last layer of the pre-trained DNN (pink layer after purple network diagram). To obtain simulated responses for new images not shown during the neural recordings (e.g., gaudy images), we pass images through the early layers of the pre-trained DNN (purple) and then through the linear mapping (pink). The rest of the simulated testbed (e.g., features from ResNet50 in blue and readout network in green) are identical to the testbed in Figure 2.

**b**. To confirm this is an appropriate simulation testbed, we first ask to what extent the features from pre-trained DNNs predict V4 responses, up to a linear mapping. We consider the recording session with the most images and repeats (1,083 natural images, each with $\geq$ 10 repeats). We fit a linear mapping from pre-trained DNN features to V4 responses and report performance as a cross-validated normalized fraction variance explained, computed in the same manner as previous studies [9, 11] (see Methods). We find that the pre-trained DNN features are highly predictive of V4 responses (mean normalized frac. var. explained: 45.1%, 54.8%, 37.7%, 56.4% for VGG19, InceptionV3, DenseNet169, and ResNet50, respectively), consistent with previous studies [11, 13].

**d**. We now use the simulation testbed in **a** to test if gaudy images improve DNN prediction with fewer images than if training on normal images. We perform the same training procedure as in Figure 2 with the same readout network architecture as in Figure 2**c** (i.e., a 3-layer convolutional neural network with relu activation functions). We find that even with this simulated testbed (where simulated responses are more likely to resemble real visual cortical responses), gaudy images improve DNN prediction with fewer training images (orange lines above black lines). This effect holds across different pre-trained DNNs (panels left to right: VGG19, InceptionV3, DenseNet169), which may extrapolate V4 responses differently to new images.

**e**. Same as **d** except for sigmoid activation functions.

**f**. Same as **d** except for a ResNet-block architecture (see Supp. Fig. 3**b**).

These results, which stem from an even more realistic testbed than that of Figure 2, provide further evidence that gaudy images improve DNN prediction with fewer training images.

Error bars in **d**-**f** indicate 1 s.d. over 5 runs (some error bars are too small to see).

a CIFAR10 example images

gaudy images lose object information, but object is still recognizable

b

CIFAR 10

CIFAR 100

c

Supplementary Figure 7: Gaudy images can be used for data augmentation.

In this paper, we find that gaudy images help to train DNNs to predict neural responses given a limited number of training images. An open question is whether gaudy images may be helpful to train DNNs for other tasks. For example, a common approach to train DNNs for object recognition is to use data augmention, which applies various transformations to a training image (e.g., a rotation or translation) while keeping the output label (i.e., the object class) fixed. A natural question is whether gaudy images can be used for data augmentation. If so, it suggests that gaudy images may be helpful in training DNNs for general, data-restricted tasks and not specialized to the task of predicting visual cortical responses.

Here, we test if gaudy images are helpful to use as a data augmention transformation for a popular object recognition task. This task, called CIFAR [76], has 50,000 training images and 10,000 test images, all labeled and $32 \times 32$ pixels. For CIFAR10, each image belongs to one of 10 classes, while for CIFAR100, each image belongs to one of 100 classes. To predict the class of an image, we train a DNN with the VGG16 architecture used in previous studies [47, 15, 46], optimized by Adam with an initial learning rate of 1e-3 and trained for 10 epochs.

**a**. Example CIFAR10 images with their corresponding labels. A key assumption of data augmentation is that performing an image transformation should not lose too much information about the object in the image. Here, we find that the gaudy versions (bottom row) of normal images (top row) do lose information about the object (e.g., color and texture) but, for the most part, still allow the object to be recognized by human eye in most cases.

**b**. We find that gaudy images is a useful transformation for data augmentation. We train the DNN on the CIFAR10 dataset (left panel) and the CIFAR100 dataset (right panel). We first train the DNN on the normal training images with no data augmentation (black line) while varying the number of training images (i.e., for 10k, we use only 10,000 normal training images, randomly selected from all possible training images). We then train the same DNN with both normal images and gaudy versions of the same normal images (orange line). For this case, a value of 10k on the x-axis indicates training on 10,000 normal images *and* 10,000 gaudy versions of the same normal images (i.e., 20,000 training images total). Performance is measured on heldout images and labels as the top-1 accuracy, which is the zero-one loss between the index of the largest softmax output of the DNN and the index of the correct class. Lines indicate mean accuracies over 5 runs with different initial seeds and samplings of training images; error bars indicate 1 s.d. over the 5 runs.

(continued on next page...)

**Supplementary Figure 7:** (...continued from previous page.)

We find that gaudy images, when used for data augmentation, increase accuracy (orange lines above black lines), especially for data-limited and difficult tasks (1k-20k training images, CIFAR100). This suggests that gaudy images may be useful for training in other data-limited tasks. We note that our focus here is not to obtain state-of-the-art performance on CIFAR (which would require other data augmentation and optimization strategies) but rather to test if gaudy images may be helpful for data augmentation.

**c**. A commonly-used data augmentation transformation is to change the percent contrast of the image, typically up to 10%-20% [75]. Because gaudy images resemble a large increase in percent contrast, we wonder if increasing the percent contrast to much larger levels than is typical (e.g., by 300%) would help or hinder DNN performance. To test this, we trained a DNN (CIFAR100 task with 10k normal training images) with augmented training data of varying levels of percent contrast. We find that all percent contrasts above 100% (i.e., no data augmentation) increase test accuracy (blue dots above black dot). This suggests that even extreme levels of some image transformations (e.g., a gaudy image or percent contrast at 400%) may be helpful for data augmentation.