[Reviews · NeurIPS 2020]

Review 1

Summary and Contributions: This paper proposes and evaluates a natural image manipulation, "gaudy" images, which reduces the number of training images needed to train DNNs. The authors evaluate the improvement from training on these image manipulations within the context of training DNNs to predict visual neural activity from the responses of another DNN to images. The authors perform analyses to find the aspects of gaudy images that make them successful, which turns out to be increasing the contrast of edges and driving diverse responses. The authors demonstrate that training with these gaudy images outperforms standard active learning algorithms.

Strengths: The paper motivates the use of gaudy images well. Instead of just defining them and showing results, they provide some intuition for why they might be useful through the derivation of the optimal best next image to pick under certain assumptions. This paper also contains very thorough and informative analyses. I especially like the efforts to understand why gaudy images are useful through specific further manipulation of the gaudy images. While many may have guessed intuitively that the high contrast edges are important, it is important to scientifically validate those intuitions. The careful comparisons to other active learning methods really underlines the usefulness of this tool Overall, gaudy images do seem like an interesting training tool for visual networks, especially in limited data regimes.

Weaknesses: The main weakness of the work to me was the context in which the authors proposed and evaluated the use of gaudy images. Modeling neural activity using DNNs is of great importance in computational neuroscience. However, the authors motivate this specific method because they claim there is not enough neural data to train complex models on normal natural images, either straight from image to neural activity or from pre-trained DNN activities to neural activity. I find both of these claims misleading: there is quite a bit of work successfully training end-to-end deep networks from images to neural predictions even with limited experimental data. I especially object to the author's explanation that linear mappings are used between pre-trained DNNs and neural activity due to lack of data. This is a large part of their motivation for the work as they use gaudy images to train another DNN. However, these linear mappings have been used for reasons more important than lack of data (if that was even a consideration at all, which I do not know). Researchers usually want to compare the type of information contained in hierarchical layers of biological vs artificial networks. Training a whole other complex model makes this question harder to answer: for example, you would have to ask does layer 2 (a CNN layer) correspond with V4 (a brain region) or are the V4 responses much more complex than those in layer 2 but are still well captured because there is an additional whole DNN transforming the features. Linear mappings aren't perfect either but at least help researchers understand whether artificial layers and brain regions represent similar features (that are linear maps of each other). Besides these specific issues with the motivation, I think the authors present gaudy images as a very useful tool beyond this specific setting, which they may be but has not been shown! I would have found the paper much stronger if they were evaluated in other settings, especially more widely used ones such as few-shot learning and so on. I would especially have found it useful if they studied DNN training directly on the gaudy images: in this modeling set-up, they are never training a DNN directly on the gaudy images but on the responses of another DNN to the gaudy images. Even within the narrow context they chose, they do not use real neural activity, just simulated, so they do not fully evaluate their model. Furthermore, it is not clear that their modeling efforts are especially useful even in a simulated version of a specific task. They find that training a DNN on a mix of gaudy and normal images improves neural prediction models with limited data. However, in at least one case linear models outperformed DNNs with limited data regardless of image type which negates the original motivation for using gaudy images. The authors point out that DNNs outperform linear models with more data, but then the gaudy images will be less useful so there seems to be a contradiction. --- UPDATE AFTER AUTHOR REBUTTAL --- Thanks to the authors for their feedback. I am not changing my score as my main concern (the limited context in which the authors present their work) remains. In their rebuttal, the authors make a compelling case for why modeling on real neural data is not always necessary for NeurIPS (which I agree with). However, I do not think some of their points apply to this specific paper. I would love to see DNN encoding model fits to real neural data using their gaudy image approach for training (vs without gaudy images). Most notably, this would not require newly collected data on gaudy images but any already existing visual neural data to natural scenes. And this would not be a new neuroscientific finding - it could just be showing improved performance over linear mappings from DNNs/using non-gaudy images (although as a side note, I think NeurIPS can be an appropriate venue for new neuroscientific findings). More importantly though, the real vs simulated neural data was not my only concern about the narrowness of the context given the claims in the paper. Using gaudy images for training additional DNNs from DNNs to neural responses is already a fairly niche context. If accepted, I encourage the authors to reduce some of their claims about the general usefulness of gaudy images since they do not test these and it is especially unclear if training DNNs directly on gaudy images (as opposed to on other DNN responses to gaudy images) will work (ex lines 16-17 "as well as aid general practitioners that seek ways to improve the training of DNNs").

Correctness: Yes, the methodology seems correct.

Clarity: The paper is very well written - I found it very easy to understand what they were doing and why.

Relation to Prior Work: As discussed above, I felt the authors misrepresented or missed some prior work related to predicting visual neural activity. Otherwise, the discussion of previous work seemed sufficient.

Reproducibility: Yes

Additional Feedback:


Review 2

Summary and Contributions: The paper investigates the added benefits of using an image preprocessing method (to increase the contrast of images such that the edges and colors in the images are overemphasized) for training DNNs to be used in modeling the activity of visual cortical neurons. Authors call the images produced by this preprocessing approach gaudy images. The expected added benefit from this approach is reduced number of training images that are required to predict neural responses. This is an important benefit since neuroscience experiments are usually rather limited in terms of the amount of data that can be collected due to practical reasons. The authors test the approach on simulated neural activity of the visual cortex, and show that using the so called gaudy images indeed improves the efficiency of training in simulations. The study provides a methodological contribution to the field. Since the presented experiments are limited to simulations, it is rather difficult to assess the degree of this contribution. I find the idea great and look forward to seeing it in practice, however at the current state, even though the simulation experiments show promising results, the idea is not sufficiently tested to demonstrate its validity and impact.

Strengths: Simple, elegant idea that is inspired by active learning in machine learning. The concept of creating a dataset made up of only the extreme values is a useful one. Interesting that this has not been done before. Experiments on simulated datasets are thorough and have very promising results.

Weaknesses: Although I really appreciate the fact that the authors chose to thoroughly investigate their idea via simulations to avoid the costs of biological experiments, I find that this manuscript is more of a preliminary study or a computational pilot study rather than a NeurIPS paper. Two major assumptions have to be made in order to be able to evaluate this study as a significant contribution. The first one is that the results from the simulation experiments would generalize to real neural data. The second one is that the gaudy versions and the original images share the same neural responses at least to a significant extent. It is impossible to say whether the first one is a correct assumption before running the actual in vivo experiments. The second assumption is also not an intuitive one to make in my opinion, and it can also only be only confirmed via in vivo experiments.

Correctness: The methodology seem to be correct but as I mentioned above, some claims rely on assumptions that can and should be tested before they can be sufficiently supported.

Clarity: Paper is very well written. Figures are beautiful and clear. Methods are well explained. Thanks for this beautiful presentation of the elegant idea.

Relation to Prior Work: Yes, I think the prior work and relation to field is sufficiently discussed. Only, I am missing a reference for active learning theory in the introduction.

Reproducibility: Yes

Additional Feedback: Edit: Thanks to the authors for their response. My primary concern was the lack of an empirical evaluation of the proposed method because simulation results might not generalize to empirical results. In response to this concern, the authors made three counter points to justify why empirical evaluation is not needed, which I did not find convincing enough to change my review/score. Below are the summary of these points and my feedback. (i) There is precedent to publish similar papers in NeurIPS -> I am not against publishing simulation results in NeurIPS per se. I am not convinced "this" paper is suitable for publishing in NeurIPS as is. (ii) This paper has to be published to convince experimentalists to use it. -> What this paper is missing is something like a simple comparison of how well a regular DNN versus one trained on "gaudy" images predict neural responses of a couple of subjects. It definitely is not the same thing as a full-blown neuroscience experiment to test a specific hypothesis, which would even make it more convincing for such a study. (iii) If they had such an empirical evaluation, they would not submit their paper to NeurIPS but a neuroscience journal. -> Some of the best neural encoding/decoding papers have been published in NeurIPS, most if not all of which have some sort of an empirical evaluation of the proposed method. If for some reason it is not possible for the same to be done for this method in a NeurIPS submission, then perhaps a neuroscience journal indeed is a better alternative. In summary, while I think that this is a good paper otherwise as I mentioned in my original review, I still find it to be borderline in the absence of some sort of empirical evaluation.


Review 3

Summary and Contributions: The authors propose a contrast-enhancing transformation of images to more efficiently train DNN-based models of the visual cortex. The transformation is simple: each pixel value is set to 0 or 255 depending on whether it is above or below the image mean. They term the resulting images “gaudy” images, because of the resulting bright, bold colors. The authors perform extensive experiments to examine the method’s effectiveness with various model architectures, and to compare to other active learning techniques.

Strengths: Image transformations that can improve the data-efficiency of neural networks are of obvious utility. This is particularly true of DNN models of visual cortex, where dataset size is limited by experimental constraints. The proposed method is fast and easy to implement. The experiments in the paper make a very convincing case for the method, at least in the simulation setup wherein visual cortex neurons are modeled as an embedding of an inner layer of a pre-trained DNN like ResNet50. The edge-only enhancement experiments give a nice interpretation of why this method may work. While data augmentation techniques like image rotation, blurring, jittering, noising, are well known, the transformation proposed here (contrast enhancement) seems to be less popular. If this is truly a general-purpose technique to speed up model training, it should become more well-known.

Weaknesses: The main weakness of this paper is that there is no proof that the findings will translate to real visual cortex data, or even outside of simulators that are based on CNNs. There are also no clear applications outside of the simulated models of visual cortex. The title of the paper is somewhat ambiguous, but on first reading, does not make this clear. I would argue that the authors should re-title the paper to use the phrase “simulated visual cortex”. Given that the simulator model and the feature extraction models are both CNNs, their findings could be due to artefacts of CNN architectures, and are not transferable to other model classes (or to real visual cortex data). It may be difficult to get real data from electrophysiological experiments to validate this method, but I believe the authors could build up the case for this method in other ways (some of which are alluded to in the discussion section): Show benefits on an end-to-end classification task. Show benefits on a data-limited transfer learning task where we have real data, where we don’t have to rely on a simulation. Somehow make the simulator more distinct from the model, so we can better understand if this is an artefact of DNNs (I consider a Gabor model to be a close cousin of a DNN). Run experiments on a larger diversity of visual cortex simulators than CNNs.

Correctness: The claims and methods are technically correct to my judgement.

Clarity: The experiments are very clear, the figures are well-made and very easy to understand. The experiments around edge enhancement provide a good interpretation of what this method is really doing. My only objection is in the clarity of the section that motivates the definition of Gaudy images. The paper states multiple times that the images can be chosen before any training data is collected. This is clearly true of gaudy images, which are based on a fixed transformation. However, the motivation from active learning for the linear model assumes that the agent makes a decision on the next datapoint given the prior covariance matrix 𝚺 (which does depend on data gathered so far). The authors state that “the optimization … does not depend on previous responses y nor the current model’s weights \beta^hat”. That said, the general thrust of this section seems clear: we should seek to increase variance along the ‘small’ principal component axes. Gaudy images increase variance for all pixels. It would be interesting to compare the PCA results of natural images and gaudy images, to see if the gaudy transformation increases variance in the lower principal components of natural images, and if there is truly a connection to the linear case.

Relation to Prior Work: The authors cite a body of work about stimulus design for neurophysiological experiments (their citations 10-15, 50, 51). However, they do not do much comparison to these methods. I would suggest that the authors implement some of these adaptive stimulus selection methods, in addition to the active learning methods implemented.

Reproducibility: Yes

Additional Feedback: *** Comments after reading author rebuttal *** I appreciate the author's responses to my review and pointers to cases where my concerns were already addressed in the supplemental material. I would agree with the authors points 1,2,3 that real electrophysiological data is not a prerequisite for a NeurIPS paper. However, the authors title the paper "High-contrast “gaudy” images improve the training of deep neural network models of visual cortex", hence I think the authors set themselves a bar to provide evidence that their results are not limited to a DNN simulator network. As other reviewers have stated, there are a few experiments the authors could perform in lieu of true electrophysiological experiments: 'gaudy-izing' images of training sets of open neural datasets, or well-known classification / transfer learning tasks. The authors do perform an interesting analysis in Supplementary Figure 6 which indicates that gaudy images could be useful for a data augmentation, but do not then actually perform a data augmentation experiment to demonstrate. The authors state "We are unaware of any other models that achieve such high performance and request R4 to provide citations to these other models to improve our paper." I would agree that DNNs are state-of-the-art in prediction accuracy, and are the best simulators of visual cortex neurons. However, my point is that the authors could strengthen their case by demonstrating that simulators of a more distinct model class from the base networks of their model, still show the gains from gaudy images. Some examples are an HMAX-type model or specifically in V4: - Cadieu, Charles, et al. "A model of V4 shape selectivity and invariance." Journal of neurophysiology 98.3 (2007): 1733-1750. - Sharpee, T.O., Kouh, M., Reynolds, J.H. Trade-off between curvature tuning and position invariance in visual area V4. (2013) Proceedings of the National Academy of Sciences of the United States of America. 110(28):11618-23. DOI: 10.1073/pnas.1217479110. DNN models of visual cortex neurons still leave a lot of unexplained variance on the table, and hence there is reason for skepticism that results on DNN simulators will translate to real data. I assert that showing the robustness of Gaudy image active learning to different simulator function classes would strengthen the authors' case, even if those simulators are not as accurate as DNNs. With respect to PCA, the authors point me to Supplementary Figure 1. This indeed exactly what I was curious about. This certainly makes stronger the link between gaudy images and active learning for the linear model. It would also be interesting to compute PCA of the activations of the base network. I also appreciate the authors clarification that, indeed, stimulus choice in the linear model can be done without looking at actual observations y, and hence can be done before running an experiment. *** End of comments (review otherwise is the same) ***


Review 4

Summary and Contributions: The authors present an class of stimuli that can increase the data-efficiency of training models of visual cortex. These stimuli are generated by binarizing the pixel values, simple transformation of the natural images used currently. These images help to improve training efficiency for simple LN models, as well as deep read-out networks. Moreover, the authors offer insight on why Gaudy images may be effective - they increasing the contrast, especially for edges. In limited settings, pre-selecting a training set of gaudy images might give better performance than some active learning methods.

Strengths: - Very clearly written paper, a lot of checks on the efficacy of the main idea has been presented in the main text and the supplement. - Each result is supported by extensive hyperparameter sweep to make the results more rigorous. - Though the paper describes the utility of gaudy images for improving training for readout networks for visual cortex, the concept can be extended in principle to general deep learning training for images. Not clear how to extend to other modalities such as text.

Weaknesses: - The analysis is restricted to simulated visual cortex responses (not real data). The simulation is based on other DNNs that are shown to have moderate (~0.6) correlation to observed neurons. However, do these simulations have reasonable accuracy for predicting responses of real neurons to gaudy images as well? Does the firing rate (or other response features) of neurons in simulation reflect the statistics of real responses? Since the ground-truth DNN was trained on low-contrast images, these models will have large firing rate in response to gaudy images (Fig 3). However, this might not be true in the real neurons due to the presence of adaptation. Due to this, the simulated cortical responses might not be reflective of responses of real neurons to gaudy images. If this is the case, the conclusions in the paper cannot be trusted. - The mismatch between training and testing distributions introduced due to Gaudy images might limit it's wide applicability for general DNN training. Even for smaller readout DNN, using gaudy images leads to a drop in final prediction accuracy (even though training is faster) - see Supp. Figure 3.

Correctness: - Line 201-202 and 220-221; "High-contrast edges are necessary and sufficient to efficiently train DNNs" -> "High-contrast edges in the gaudy images are necessary and sufficient to efficiently train DNNs". The analyses have shown a weaker point.. - The distribution of natural image pixel intensities is not symmetric. The mean is darker, with a heavy tail. Does it matter if we binarize with mean, or other quantity, such as the median? - Most other analyses are well described and thoroughly analysed.

Clarity: - At various places, the authors refer to general "increases in performance" (Ex. line 153, 204) .. they could be replaced with more accurate terms (training data efficiency, final accuracy, etc).. - If scaling the image by 400% gives same performance as gaudy, then why not just do that? Scaling gives a flexible data augmentation scheme that, where you can sample different image scales (say, between 100% - 600%) and perhaps get similar improvement.. the extra flexibility might also help in better addressing the distribution mismatch problem that arises with gaudy images.. Addressing how gaudy images are different than this more straightforward method might be useful.

Relation to Prior Work: - Note that binarizing the visual stimulus is a common practise for white noise stimulus. Experimenters routinely use a white noise stimulus where each pixel is 0 or 255 (for a 0-255 display) rather than sampling from a gaussian distribution. This increases firing rate of the cells, and gives receptive field estimates very quickly. This work is clearly novel in the context of DNN, but connection to this standard practise should be made. - make a separate prior works section.

Reproducibility: Yes

Additional Feedback: Broader impact: Please include the fact that reducing the data requirement will make the animal experiments shorter, and potentially reduce animal suffering.

[Author Response · NeurIPS 2020]

We thank the reviewers for their thorough and constructive reviews. We first address a concern raised by all reviewers and then respond to each reviewer individually.

**Application to neural recordings:** Although all reviewers found the work interesting, and the simulations realistic and compelling, a shared concern was that we have not yet tested gaudy images with real neural data collected from macaque V4. We are eager to try these experiments, but we respecfully submit that requiring an application to neural data is too high a bar for the following reasons:

1. There is strong precedent in NeurIPS and the computational neuroscience field to perform realistic simulations on new methods/approaches that then inspire new neuroscientific experiments (see [12, 14, 48-55]). Almost all NeurIPS studies rely on either simulations or previously-collected data to validate their approaches/methods. Our paper is no different, and we use state-of-the-art models and realistic paradigms. Our work goes beyond simulations, as we propose a modeling framework (i.e., a readout network) and provide intuition about DNN models that can be used for future experiments and models (e.g., DNN responses are strongly driven by high-contrast edges).
2. The reviews highlight a chicken-and-egg problem: We wrote this paper to convince experimentalists to run these costly experiments, but the reviewers suggest we should have already collected this data to convince experimentalists. We believe NeurIPS is an appropriate place for our work, as we propose a proof-of-concept, backed by thorough and realistic simulations, which will inspire experimenters and machine learning researchers to test the efficacy of gaudy images in their own work.
3. If we had already collected neural data, we almost certainly would not submit this work to NeurIPS, which is not an appropriate venue for new neuroscientific findings. This is because NeurIPS does not have ethical requirements for the treatment of animals nor statistical requirements for number of subjects, etc. In addition, due to the 8-page limit of NeurIPS, we would need to include experimental details—critical for assessing neuroscientific claims—as Supplemental Material, which is optional and thus not guaranteed to be peer-reviewed for NeurIPS.

For these reasons, we believe that our work advances the field and should not be penalized for not performing new neuroscientific experiments. **We respectfully ask reviewers to increase their score, if they agree.**

We will update the text with all of the reviewer's comments. We respond to a few of those comments here:

**R1:** We agree about linear mappings and will provide broader scope about them. We are unaware of any successful end-to-end DNNs to predict V4/IT responses (although we cite two for V1 responses [8,9]) and request R1 provides citations for them to improve our paper. Although R1 finds that a linear model outperforms the 3-layer relu network, it does not outperform the sigmoid or Resnet-like networks. This suggests gaudy images are more useful for models that are good fits to the ground truth responses.

**R3:** re: results from simulations would generalize to real neural data. We base this assumption on many previous studies that find DNNs are appropriate models for neural responses [3-10,13-16,23] and that our results generalize across multiple DNN models. Thus, we have provided strong evidence to convince experimenters to collect real data. We feel this is the epitome of a NeurIPS paper: Propose a computational approach/method and demonstrate its effectiveness in order to inspire new experiments. re: "gaudy versions and the original images share the same neural responses at least to a significant extent". We disagree with R2 that we make this assumption. This assumption is more in line with data augmentation, which transforms images but assumes the labels remain the same. Here, gaudy images drive *different* responses than normal images (Fig. 3a). We believe this will hold for neural data and is one of the reasons why gaudy images train DNNs more efficiently.

**R4:** re: "no clear applications outside...visual cortex". We provide evidence that gaudy images may be useful for data augmentation for object recognition (Supp. Fig. 6). Please see our comments to all reviewers. re: other models of visual cortex to assess gaudy images. To our knowledge, for our simulations we have used models that achieve the highest predictive performance for neural data [7]. We are unaware of any other models that achieve such high performance and request R4 to provide citations to these other models to improve our paper. re: active learning of the linear model. We note that $\Sigma = \mathbf{X}\mathbf{X}^T$ is the covariance matrix of images $\mathbf{X}$ and does not depend on responses $y$. Thus, all images can be chosen before recording any responses. re: PCA on normal vs. gaudy images. We have applied PCA to the pixels (Supp. Fig. 1a-c) and find gaudy images do increase the variance for the lower PCs. re: other adaptive stimulus techniques. These methods are not applicable to our setting. Some methods do not consider training any model [13-16,50] and some methods consider models with $< 1k$ params [11,12,51,52] (we have 1.5 million params).

**R5:** re: "drop in final prediction accuracy". Because AL algorithms explicitly mismatch the training distr. from the test distr., this drop is expected from any AL algorithm when sample size grows large (when the prior of the AL algo becomes too strong). We propose a hybrid approach for large-data regimes (Supp. Fig. 3a caption). re: mean vs. median vs. contrast. We have found that all of these yield similar performance (including binarizing based on each channel separately), so we chose the simplest (i.e., thresholding on the mean). We will include these results. We note that one can scale gaudy images just as one would scale contrast.

[Meta-Review · NeurIPS 2020]

This paper had borderline scores. Overall, I think this paper presents a nice core finding that was sufficiently well validated in the context of simulations. The simulated results are reasonably compelling and relatively thorough analyses were presented. In the discussion with reviewers, there was a reasonable consensus that the author response overemphasized the extent to which the reviewers were hung up on the lack of experimental data. While R3 felt most strongly that this specific paper was not strong enough without further empirical validation, this was clarified to not be a bias against simulation papers generally, but rather the reviewer's opinion that they were not convinced this specific paper's results were strong enough without real data. Other reviewers had indicated that their concern was actually, at least in part, that the authors simply overstated their claims given the lack of empirical validation (more below). There was also a sense by R1 and R4 that perhaps the authors might have been able to have found a way to leverage pre-recorded neural data (collected under conventional settings) to somehow partially validate the claims of the paper. The author response did not attempt to address this point, and this was dissatisfying to the reviewers. While at this point I don't believe I'm capable of further enforcing changes to the paper, there are a few summary points which warrant further attention. Remaining concerns 1) Lack of nuance in motivation: It is unambiguously clear that it would be useful to reduce the amount of experimental time required to directly fit models of neural responses. This was part of the motivation, and it is essentially correct. Nevertheless, this work does not fully contextualize itself relative to attempts to perform direct fitting and instead just asserts that it isn't presently tractable. There are, as R1 noted, multiple previous papers which successfully train direct models of neural activity from relatively small quantities of data, as well as other methods that have attempted to come up with creative approaches around this problem. For example, multiple papers by Fetz and colleagues in the early 90s did versions of direct training using RNNs to predict neurons in the motor system, admittedly for small input spaces. More recently, there has been some work directly fitting neurons in the retina with relatively small amounts of data (e.g. McIntosh et al. 2016, Batty et al. 2017). And for visual cortical neurons, papers from the Zylberberg lab are particularly relevant. Specifically, Kindel et al 2019 performs direct fitting from publicly available data. Confusingly, the authors cite this paper along with a large group of references in support of the statement "To date, the most successful applications of this approach use visual features extracted from natural images by pre-trained models". However, as far as I can tell, Kindel et al in fact comes to the opposite conclusion from what was stated in the present submission. In that work, a headline result was that fully data driven model of V1 outperformed pretrained models (Fig 4). I found the misrepresentation of this work particularly alarming since it is a work about V1 and the title, abstract, and framing of the present submission focus on visual cortex. While the authors were correct to point out in their response that direct training by neural networks specifically of V4/IT neurons has probably not been done, those regions were not well framed as the exclusive focus of the present submission. And if those brain regions are the main focus, other work by e.g. Pasupathy and Connor or Cadieu et al (mentioned by R4) would be relevant to discuss. Overall, I believe the authors could have situated their work better...that is, without errors and with more nuance. In my reading of the reviews, this was core to R1's concerns and I share them. "Improved object recognition using neural networks trained to mimic the brain’s statistical properties" Federerer et al 2020 (published after NeurIPS submission, but possibly of interest) "Multilayer recurrent network models of primate retinal ganglion cell responses" Batty et al. 2017 "Using deep learning to probe the neural code for images in primary visual cortex" Kindel et al 2019 "Deep learning models of the retinal response to natural scenes" McIntosh et al. 2016 2) The use of the word "gaudy" to describe the images used, while perhaps meant to be catchy, seems to me very misleading, given that this word already has a common usage that doesn't seem to me to align very well with the proposed technical definition. I asked the reviewers what how they felt about this point. Two reviewers agreed this word choice is confusing (the other two did not address this point). The work seemed to me to simply be using a kind of "saturated" image. Suggestions by reviewers for alternatives might include sticking with "high-contrast" or using either "color quantized" or "binarized natural images". 3) The title and some of the claims seem to overreach, as discussed above. In the reviewer discussion and responses it was clear there was a bit of irritation that the title and some comments within the paper implied stronger validation than existed in the actual paper (as R4 noted, with a suggestion to rephrase to “simulated visual cortex”). Again the authors responded that it was too much to ask for them to perform electrophysiology experiments before publication in NeurIPS. While three out of four reviewers and this AC were willing to accept this claim in part, what was primarily being argued was that the paper should more accurately reflect its limitations and caveats, especially around the fact that it hadn't performed real-world experiments. I would strongly urge the authors to address these summary points, as well as specific comments by the individual reviewers in their revisions. Nevertheless, the quality of the core contributions along with the generally clear presentation and strong simulated validations make me comfortable recommending this paper for acceptance.